# Measuring Representational Shifts in Continual Learning: A Linear Transformation Perspective

**Joonkyu Kim** [1]   **Yejin Kim** [2]   **Jy-yong Sohn** [2]

## Abstract

In continual learning scenarios, catastrophic forgetting of previously learned tasks is a critical issue, making it essential to effectively measure such forgetting. Recently, there has been growing interest in focusing on *representation forgetting*, the forgetting measured at the hidden layer. In this paper, we provide the first theoretical analysis of representation forgetting and use this analysis to better understand the behavior of continual learning. First, we introduce a new metric called *representation discrepancy*, which measures the difference between representation spaces constructed by two snapshots of a model trained through continual learning. We demonstrate that our proposed metric serves as an effective surrogate for the representation forgetting while remaining analytically tractable. Second, through mathematical analysis of our metric, we derive several key findings about the dynamics of representation forgetting: the forgetting occurs *more rapidly* to a *higher* degree as the layer index increases, while increasing the width of the network slows down the forgetting process. Third, we support our theoretical findings through experiments on real image datasets, including Split-CIFAR100 and ImageNet1K.

## 1. Introduction

Continual learning, referred to as lifelong learning, involves training models on a sequence of tasks with the aim of achieving good performance for all tasks (Chen & Liu, 2018; Hadsell et al., 2020; Kudithipudi et al., 2022). This framework is particularly compelling as it closely mimics the human learning process; an individual continually acquires new skills and knowledge throughout their lifetime. However, unlike humans, who can retain and recall a good amount of memories from the past, the trained models experience a significant decline in performance on the previous tasks as they adapt to the new ones. This phenomenon, known as catastrophic forgetting, is described as the model losing *knowledge* of the previous tasks and is typically evaluated by measuring the drop in accuracy on the previous tasks (McClelland et al., 1995; McCloskey & Cohen, 1989).

Internally, the representations of the model, that is, the hidden activations, undergo a drastic shift during continual learning as well (Caccia et al., 2021). This calls for a separate term that indicates the loss of knowledge for previous tasks in terms of the *hidden representations*. In this paper, we will use the term *representation forgetting of task $t$* to refer to the degradation in a model's ability to produce high-quality features for task $t$. Note that we are not the first to use the term *representation forgetting*, as it has been used in the literature to indicate representational shifts during the continual learning process in general (Davari et al., 2022; Luo et al., 2023; Zhang et al., 2022; Murata et al., 2020).

Understanding the behavior of the representation forgetting, which is measured at the level of the intermediate layers, is equally important to that of the observed forgetting (forgetting measured through the model's final output), because (1) there are situations such as in unsupervised continual learning (Rao et al., 2019; Fini et al., 2022) where our interest is in the inner representations of the model rather than the model's output, and (2) understanding the representation forgetting itself can lead to better understanding of the observed forgetting.

Although there have been several empirical results on the representation forgetting under continual learning (Caccia et al., 2021; Davari et al., 2022; Hess et al., 2023; Luo et al., 2023), no clear justification was provided in terms of theory. Our work addresses this gap by offering one of the first theoretical analyses of the representation forgetting. To achieve this, we introduce a novel metric, *representation discrepancy*, which quantifies the minimum alignment error of two representation spaces under a linear transformation. We show that this metric not only effectively measures rep-

[1]Department of Electrical & Electronic Engineering, Yonsei University, Seoul, South Korea [2]Department of Statistics and Data Science, Yonsei University, Seoul, South Korea. Correspondence to: Jy-yong Sohn <jysohn1108@yonsei.ac.kr>.

*Proceedings of the $42^{nd}$ International Conference on Machine Learning*, Vancouver, Canada. PMLR 267, 2025. Copyright 2025 by the author(s).

resentation forgetting, but also enables analytical feasibility.

Using the representation discrepancy, we focus on comparing the representations of two models $h_t$ and $h_{t+\Delta t}$, where $h_t$ represents the model trained incrementally up to task $t$, and analyze how the representation evolves over the continual learning process. Our main contributions are summarized as below.

**Main Contributions:**

- We propose a novel metric *representation discrepancy*, denoted by $D_t^k(h_t, \Delta t)$, which measures the discrepancy of the representation spaces of models $h_t$ and $h_{t+\Delta t}$ (for task $t$ at layer $k$), when an optimal linear transformation $\boldsymbol{T}$ is applied to align the spaces. We show that $D_t^k(h_t, \Delta t)$ serves as an effective surrogate for the representation forgetting of task $t$ while also being analytically feasible.

- We construct an upper bound $U_t^k(\Delta t)$ on $D_t^k(h_t, \Delta t)$ using the theoretical framework proposed by Guha & Lakshman (2024) and derive three main theoretical findings. First, $U_t^k(\Delta t)$ exhibits two distinct phases: the *forgetting phase*, during which it monotonically increases with $\Delta t$ (the number of tasks learned after task $t$), and the *saturation phase*, where it gradually converges to the asymptotic value. Second, the asymptotic representation discrepancy $U_{t,\infty}^k := \lim_{\Delta t \to \infty} U_t^k(\Delta t)$ is linearly proportional to the size[1] of the representation space. Third, $U_t^k(\Delta t)$ enters the saturation phase more quickly in the upper layers (*i.e.,* higher $k$) compared to the lower layers, while increasing the width of the model delays the transition to the saturation phase.

- We support our theoretical findings through experiments on real image datasets, including Split-CIFAR100 and ImageNet1K. Furthermore, we empirically observe a strong linear relationship between the layer index $k$ and the size of the representation space. Combined with our second theoretical result shown above, we conclude that the asymptotic representation discrepancy increases linearly with the layer index $k$.

## 2. Related Works

### 2.1. Continual Learning Theory

Several prior works have conducted theoretical analyses of continual learning. These works can be categorized according to the modeling of the continual learning setup and their analytical framework. A simple yet effective approach

assumes the model to be linear, with each task framed as a regression problem (Ding et al., 2024; Zhao et al., 2024; Evron et al., 2022; Lin et al., 2023; Li et al., 2023; Heckel, 2022). Here, the drift in the weights during the continual learning process is generally used as a measure of the forgetting or the generalization error. For instance, Evron et al. (2022); Lin et al. (2023); Ding et al. (2024) examines how task sequence properties (*e.g.,* number of tasks, task ordering, task similarity) influence the error. Meanwhile, Heckel (2022); Li et al. (2023); Zhao et al. (2024) focuses more on the impact of regularization on the error. Another line of work employs the well-established theoretical teacher-student framework (Saad & Solla, 1995b;a; Yoshida & Okada, 2019; Goldt et al., 2019; Straat et al., 2022) to study continual learning (Lee et al., 2021; Asanuma et al., 2021). In these works, each teacher network represents a distinct task, and the similarity between teacher networks is used as a proxy for task similarity. These studies further derive connections between task similarity and the generalization error of the student network. Other works leverage the Neural Tangent Kernel (NTK) regime, where the continual learning setup is modeled as a recursive kernel regression (Bennani et al., 2020; Doan et al., 2021; Yin et al., 2020; Karakida & Akaho, 2021).

The most relevant to our work is Guha & Lakshman (2024), where they proposed using perturbation analysis to derive error bounds on catastrophic forgetting. Specifically, they viewed each learning step in continual learning as a perturbation of the weights of the model and analyzed how this affected the overall shift in the model's output on the previous tasks. By using the maximum shifted distance of the output on the previous tasks as a proxy for measuring forgetting, they established relationships between the model's depth and width to the upper bound on the forgetting error.

Our work is similar to Guha & Lakshman (2024) as we also view the continual learning process as a series of perturbation on the weights. However, the key distinction between our work and Guha & Lakshman (2024) (or in fact any other works we mention here) lies in our focus on measuring the forgetting of the *representations* for the previous task. Simply measuring the drift in the activations or the hidden layer weights of the model during the continual learning process fails to fully capture the representation forgetting, as the internal features may change but end up producing the same output. This leads us to propose a novel metric, the *representation discrepancy* $D_t^k(h_t, \Delta t)$, which quantifies the minimum alignment error between two representation spaces under a linear transformation. Note that one should not confuse *representation discrepancy* with *discrepancy distance* introduced in Mansour et al. (2009). Discrepancy distance refers to the distance in terms of data distributions whereas representation discrepancy refers to the distance in terms of the representation spaces. A formal definition of

---

[1]The size of the $k$-th layer representation space of task $t$ is the norm of its biggest feature, as defined in Def. 2.

$D_t^k(h_t, \Delta t)$ is provided in Def. 4.

Other popular approaches include leveraging the PAC-Bayesian framework to establish error bounds for future tasks based on observed losses on current tasks (Pentina & Lampert, 2014) or to derive lower bounds on the memory requirements in continual learning (Chen et al., 2022). Additionally, some works focus on studying the continual learning setup itself, decomposing the problem into within-task prediction and out-of-distribution detection (Kim et al., 2022), or connecting it with other problems such as bi-level optimization or multi-task learning under specific assumptions (Peng et al., 2023). Researchers have also explored sample complexity (Cao et al., 2022; Li et al., 2022), the impact of pruning (Andle & Yasaei Sekeh, 2022), and the effects of contrastive loss on generalization (Wen et al., 2024) in the continual learning regime.

## 2.2. Representation Forgetting

The concept of representation forgetting in continual learning is not yet well-established in the literature, leading to the introduction of various measurements. The most relevant to our work is Davari et al. (2022), where they propose to use the drop in linear probing accuracy as a measure of the representation forgetting. Similarly, Zhang et al. (2022) devises a protocol for evaluating representations in continual learning, which involves computing the average accuracy of the model's features across all previous tasks. Other metrics, such as canonical correlation analysis and centered kernel alignment are widely used to compare the representation spaces between models (Kornblith et al., 2019). Ramasesh et al. (2020) employ these metrics to empirically demonstrate that catastrophic forgetting arises primarily from higher-layer representations near the output layer, where sequential training erases earlier task subspaces.

Additionally, several studies have explored representation forgetting in different continual learning setups. For instance, Caccia et al. (2021) and (Zhang et al., 2024) investigates representation forgetting in either online or unsupervised continual learning scenarios, while Luo et al. (2023) and Hess et al. (2023) investigates this phenomenon in the context of Natural Language Processing (NLP) models. Specifically, Luo et al. (2023) introduces three metrics–overall generality destruction, syntactic knowledge forgetting, and semantic knowledge forgetting–to assess different aspects of forgetting for NLP models.

It should be noted that all these studies primarily focus on the *empirical analysis* of representation forgetting in continual learning, without presenting a corresponding theoretical framework. To the best of our knowledge, our work provides a theoretical analysis of representation forgetting for the first time.

## 3. Problem Setup

We consider the continual learning setup where we consecutively learn $N$ tasks, and the dataset for each task is denoted by $\mathcal{D}_1, \cdots, \mathcal{D}_N$. The continual learner is assumed to be an $L$-layer ReLU network, which is trained sequentially in the increasing order: from $\mathcal{D}_1$ to $\mathcal{D}_N$. Each task $t$ involves supervised learning on $\mathcal{D}_t = \{(\boldsymbol{x}_t^i, \boldsymbol{y}_t^i)\}_{i=1}^{n_t}$ having $n_t$ samples with features $\boldsymbol{x}_t^i \in \mathbb{R}^{d_x}$ and labels $\boldsymbol{y}_t^i \in \mathbb{R}^{d_y}$. We define $X_t = \{\boldsymbol{x}_t^i\}_{i=1}^{n_t}$ and $Y_t = \{\boldsymbol{y}_t^i\}_{i=1}^{n_t}$. The model trained up to task $t$ is denoted as $h_t$ and will take on the form

$$h_t(\boldsymbol{x}) = \boldsymbol{W}_t^L \phi(\boldsymbol{W}_t^{L-1} \cdots \boldsymbol{W}_t^2 \phi(\boldsymbol{W}_t^1(\boldsymbol{x}))),$$

where $\phi$ represents the ReLU activation and $\boldsymbol{W}_t^k$ represents the weight matrix at layer $k$. The width of the $k$-th hidden layer is denoted by $w_k$. Thus, the size of each weight matrix is as follows: $\boldsymbol{W}^1 \in \mathbb{R}^{w_1 \times d_x}$ and $\boldsymbol{W}^L \in \mathbb{R}^{d_y \times w_{L-1}}$ while the rest of the hidden weight matrix has size $\boldsymbol{W}^k \in \mathbb{R}^{w_k \times w_{k-1}}$ for $k \in \{2, 3, \cdots, L-1\}$. We mainly use superscript $k \in [L]$ to denote the layer index, and subscript $t \in [N]$ to denote the task, where we use the notation $[n] := \{1, \cdots, n\}$ throughout the paper. In addition, since we will be exclusively dealing with the intermediate layers of the model, we use $h_t^k(\boldsymbol{x})$ to denote the output of the $k$-th layer of $h_t$, when the input to the model is $\boldsymbol{x}$. Note that $h_t^L(\boldsymbol{x}) = h_t(\boldsymbol{x})$ holds.

Now, we formally define the representation space for task $t$, along with the notions of *size* and *distance* of the representations spaces. Note that this will clarify the ambiguity of the word *representation space* while also helping us to gain a more intuitive understanding of our theorems.

**Definition 1** (Representation Space for Task $t$). *For a given model $h_{t'}$ trained incrementally up to task $t'$, the $k$-th layer representation space of $h_{t'}$ for task $t$ is defined as*

$$\mathcal{R}_t^k(h_{t'}) := \{h_{t'}^k(\boldsymbol{x}) : \boldsymbol{x} \in X_t\}.$$

**Definition 2** (Size of Representation Space). *The size of the $k$-th layer representation space of model $h_{t'}$ for task $t$ is defined as*

$$\|\mathcal{R}_t^k(h_{t'})\| := \max\{\|h_{t'}^k(\boldsymbol{x})\|_2 : \boldsymbol{x} \in X_t\}.$$

**Definition 3** (Distance between Representation Spaces). *Given task index $t$, layer index $k$, and models $h_{t_1}$ and $h_{t_2}$, the distance between representation spaces $R_t^k(h_{t_1})$ and $R_t^k(h_{t_2})$ is defined as*

$$d(\mathcal{R}_t^k(h_{t_1}), \mathcal{R}_t^k(h_{t_2})) := \\ \max\{\|h_{t_1}^k(\boldsymbol{x}) - h_{t_2}^k(\boldsymbol{x})\|_2 : \boldsymbol{x} \in X_t\}.$$

Intuitively, the size of the representation space $\mathcal{R}_t^k(h_{t'})$ is determined by the norm of the biggest feature in $\mathcal{R}_t^k(h_{t'})$. Furthermore, the *distance* between two representation spaces

$\mathcal{R}_t^k(h_{t_1})$ and $\mathcal{R}_t^k(h_{t_2})$ is measured by the distance between the most distant representations for the same input $\boldsymbol{x}$, where the maximization is taken over the $t$-th task dataset.

To provide a glimpse into our approach to analyzing the representation forgetting, we offer a remark on the relationship between the metric *continual learning error* used in a related work (Guha & Lakshman, 2024) and our definition of the *distance* between two representation spaces.

**Remark 1.** *Guha & Lakshman (2024) use*

$$d(\mathcal{R}_t^L(h_{t_1}), \mathcal{R}_t^L(h_{t_2})) :=$$
$$\max\{\|h_{t_1}^L(\boldsymbol{x}) - h_{t_2}^L(\boldsymbol{x})\|_2 : \boldsymbol{x} \in X_t\} \quad (1)$$

*as a surrogate for the forgetting of task $t$ measured for the model $h_{t+\Delta t}$ trained on additional $\Delta t$ tasks. The underlying idea of using equation 1 in (Guha & Lakshman, 2024) is that, if the output of $h_{t+\Delta t}$ is close to $h_t$ for all $\boldsymbol{x} \in X_t$, then $h_{t+\Delta t}$ would perform well on task $t$. We will use a similar approach, but we measure the discrepancy of the representation space at hidden layers $k$ instead of the final output layer $L$; note that equation 1 is a special case of Def. 3 when we set $k = L$.*

## 4. Proposed Metric

In this section, we propose the representation discrepancy, denoted as $D_t^k(h_t, \Delta t)$, which measures the discrepancy of the $k$-th layer representation spaces of models $h_t$ and $h_{t+\Delta t}$ (before/after learning additional $\Delta t$ tasks) for task $t$. Note that *discrepancy* between two representation spaces is measured as the *distance* between them after aligned through a linear transformation. We use $D_t^k(h_t, \Delta t)$ as a proxy to the forgetting of representations for task $t$ over the continual leaning process, and analyze it in the upcoming sections. We first provide the motivation for such definition in Sec. 4.1 and then provide a formal definition in Sec. 4.2.

### 4.1. Motivation

As the model incrementally learns from task 1 to task $N$, the learner gets a total of $N$ models, denoted by $h_1, \cdots, h_N$. Our goal is to mathematically analyze the extent to which the model $h_{t+\Delta t}$ (learned up to task $t + \Delta t$), forgets the representations useful for conducting task $t$, where $\Delta t > 0$.

Note that numerous studies have proposed different methods to quantify the representation forgetting. One famous approach, introduced in Davari et al. (2022), measures the representation forgetting in terms of the degradation of the performance of the linear probing method. Specifically, to assess the representation forgetting at the $k$-th hidden layer of model $h_{t+\Delta t}$ on task $t$, one compares the performances of optimal linear classifiers $\boldsymbol{C}^k \in \mathbb{R}^{d_y \times w_k}$ trained on dataset $\mathcal{D}_t$ when placed on top of $h_t^k$ and $h_{t+\Delta t}^k$, respec-

tively. Formally, this would be evaluating the performance difference

$$\Delta P_t^k(\Delta t) := P(\boldsymbol{C}_t^k \circ h_t^k(\boldsymbol{X}_t), \boldsymbol{Y}_t)$$
$$- P(\boldsymbol{C}_{t+\Delta t}^k \circ h_{t+\Delta t}^k(\boldsymbol{X}_t), \boldsymbol{Y}_t), \quad (2)$$

where $P$ denotes a chosen performance metric, e.g., accuracy, and $\boldsymbol{C}_t^k = \underset{\boldsymbol{C}^k \in \mathbb{R}^{d_y \times w_k}}{\arg\min} \mathcal{L}(\boldsymbol{C}^k \circ h_t^k(\boldsymbol{X}_t), \boldsymbol{Y}_t)$ and $\boldsymbol{C}_{t+\Delta t}^k = \underset{\boldsymbol{C}^k \in \mathbb{R}^{d_y \times w_k}}{\arg\min} \mathcal{L}(\boldsymbol{C}^k \circ h_{t+\Delta t}^k(\boldsymbol{X}_t), \boldsymbol{Y}_t)$ are the optimal linear classifiers (for $h_t^k$ and $h_{t+\Delta t}^k$, respectively) with respect to loss function $\mathcal{L}$.

Note that mathematically analyzing Eqn. 2, which requires analyzing the performances of complex multi-layer neural networks, is challenging. A practical alternative inspired by the metric proposed by Guha & Lakshman (2024) shown in Eqn. 1, involves using the maximum distance between the outputs of the two models, written as

$$\max_{\boldsymbol{x} \in X_t} \|\boldsymbol{C}_t^k \circ h_t^k(\boldsymbol{x}) - \boldsymbol{C}_{t+\Delta t}^k \circ h_{t+\Delta t}^k(\boldsymbol{x})\|_2, \quad (3)$$

as a proxy for the representation forgetting.

While the metric in Eqn. 3 avoids direct evaluation of the performance $P$, it still requires computing the optimal classifiers $\boldsymbol{C}_t^k$ and $\boldsymbol{C}_{t+\Delta t}^k$. However, deriving these classifiers is nontrivial. Thus, there is a need for a metric that not only serves as an effective surrogate for representation forgetting but is also analytically feasible. This motivates our proposed metric, formally defined as below.

### 4.2. Representation Discrepancy

Below we propose a novel metric, the *representation discrepancy*, as a practical surrogate for the representation forgetting of models trained in the continual learning setup. See Fig. 1 for the illustration of our proposed metric.

**Definition 4** (Representation Discrepancy). *Let $h(\boldsymbol{x}) = \boldsymbol{W}^L \phi(\boldsymbol{W}^{L-1} \cdots \phi(\boldsymbol{W}^1 \boldsymbol{x}))$ be an $L$-layer ReLU Network with each hidden layer $k$ having width $w_k$. Suppose $h$ is trained sequentially on the datasets $\mathcal{D}_1, \cdots, \mathcal{D}_N$ and $h_t$ represents the model trained up to $\mathcal{D}_t$. For a given target task $t$, the k-th layer representation discrepancy for model $h_t$ after trained on additional $\Delta t$ tasks, is defined as*

$$D_t^k(h_t, \Delta t) := \min_{\boldsymbol{T}} d(\mathcal{R}_t^k(h_t), \boldsymbol{T}(\mathcal{R}_t^k(h_{t+\Delta t}))),$$

*where $\boldsymbol{T} \in \mathbb{R}^{w_k \times w_k}$ applies a linear transformation to all the features in $\mathcal{R}_t^k(h_{t+\Delta t})$.*

Intuitively, at layer $k$, our metric $D_t^k(h_t, \Delta t)$ measures the minimum worst-case mis-alignment between the $k$-th layer representation spaces of $h_t$ and $h_{t+\Delta t}$, where the minimization is over linear transformation $\boldsymbol{T}$ and the maximization is over samples in task $t$.

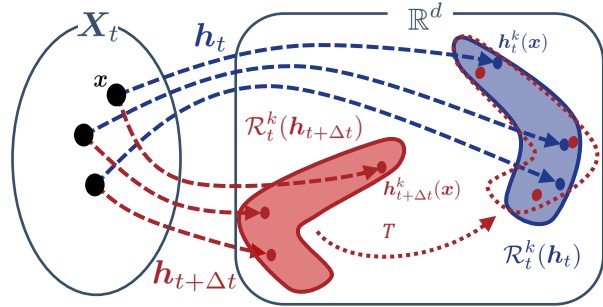

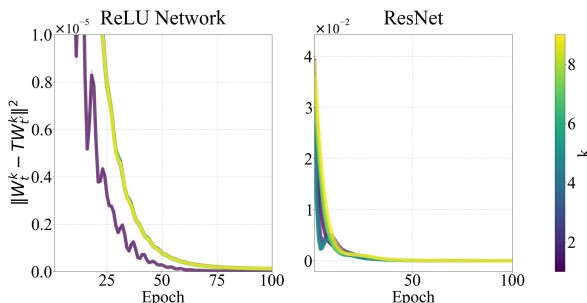

*Figure 1.* Visual interpretation of the representation discrepancy $D_t^k(h_t, \Delta t) = \min_{\boldsymbol{T}} d(\mathcal{R}_t^k(h_t), \boldsymbol{T}(\mathcal{R}_t^k(h_{t+\Delta t})))$ defined by us in Def. 4. Here, the linear transformation $\boldsymbol{T}$ is optimized to reduce the distance between the representation space $\mathcal{R}_t^k(h_t) = \{h_t^k(\boldsymbol{x})\}_{\boldsymbol{x} \in X_t}$ and $\mathcal{R}_t^k(h_{t+\Delta t}) = \{h_{t+\Delta t}^k(\boldsymbol{x})\}_{\boldsymbol{x} \in X_t}$.

We argue that $D_t^k(h_t, \Delta t)$ serves as an effective surrogate for the representation forgetting, following the logic stated in Remark 1. Note that when $D_t^k(h_t, \Delta t)$ is small, there exists a linear transformation $\boldsymbol{T}^* \in \mathbb{R}^{w_k \times w_k}$ such that

$$h_t^k(\boldsymbol{x}) \approx \boldsymbol{T}^* \circ h_{t+\Delta t}^k(\boldsymbol{x}), \quad \forall \boldsymbol{x} \in X_t.$$

Consequently, the accuracy of the linear probing on $h_t^k$ can be matched by that of the linear probing on $h_{t+\Delta t}^k$, since for any linear classifier $\boldsymbol{C}_1 \in \mathbb{R}^{d_y \times w_k}$, we have

$$\boldsymbol{C}_1 \circ h_t^k \approx \boldsymbol{C}_1 \circ (\boldsymbol{T}^* \circ h_{t+\Delta t}^k) = \boldsymbol{C}_2 \circ h_{t+\Delta t}^k$$

for some $\boldsymbol{C}_2 \in \mathbb{R}^{d_y \times w_k}$. Therefore, the lower the representation discrepancy $D_t^k(h_t, \Delta t)$, the lower the representation forgetting. Equivalently, if the model undergoes high representation forgetting, then it has a high value of $D_t^k(h_t, \Delta t)$. Our empirical observations also show that there is a strong linear relationship between the representation forgetting and the representation discrepancy. See Fig. 8 in the Appendix.

## 5. Theory

In this section, we analyze the representation discrepancy $D_t^k(h_t, \Delta t)$ in Def. 4. First, in Sec. 5.1, we provide an upper bound $U_t^k(\Delta t)$ on $D_t^k(h_t, \Delta t)$, as stated in Thm. 1. Then, in Sec. 5.2, we analyze the behavior of the upper bound $U_t^k(\Delta t)$, which gives us insights on the representation forgetting, including (1) the amount of representation forgetting in the asymptotic case (*i.e.,* when $\Delta t \to \infty$) and (2) the rate of the representation forgetting.

### 5.1. Bounding the Representation Discrepancy

Before stating our main result on bounding the representation discrepancy $D_t^k(h_t, \Delta t)$, we start with an assumption we introduced for ease of analysis, which is empirically supported in our observation in Fig. 2.

**Assumption 1.** *Let $h(\boldsymbol{x})$ be a randomly initialized $L$-layer ReLU Network with each hidden layer $k$ having the same*

*Figure 2.* An empirical result that supports Assumption 1, which is tested on two networks (a fully connected ReLU network and a ResNet) using real image datasets (CIFAR100 and ImageNet, respectively). When $t = 1$ and $t' = N$, we plot how the difference of $\boldsymbol{W}_t^k$ and $\boldsymbol{T}\boldsymbol{W}_{t'}^k$ varies as we train a linear transformation $\boldsymbol{T}$ for 100 epochs, for each $k \in \{1, \cdots, 9\}$. Note that the plot is averaged across 50 random seeds. The results show $\|\boldsymbol{W}_t^k - \boldsymbol{T}\boldsymbol{W}_{t'}^k\|_2^2$ converges to zero. This implies that there exists a linear transformation $\boldsymbol{T}$ that closely approximates two weight matrices of each layers, demonstrating that Assumption 1 holds in practice.

*width $m$. Suppose $h(\boldsymbol{x})$ is trained sequentially on the datasets $\mathcal{D}_1, \cdots, \mathcal{D}_N$ and let $\boldsymbol{W}_t^k$ represent the weight matrix of layer $k$ after learning task $t$. For each $k \in [L]$ and for any task indices $t, t' \in [N]$ satisfying $t < t'$, we assume that there exists a linear transformation $\boldsymbol{T} \in \mathbb{R}^{w_k \times w_k}$ such that the following holds:*

$$\boldsymbol{T}\boldsymbol{W}_{t'}^k = \boldsymbol{W}_t^k. \tag{4}$$

This assumption implies that for a randomly initialized $L$-layer ReLU network updated through the continual learning process, the weight matrices of layer $k$ at two different task index $t, t'$ can be aligned through a linear transformation $\boldsymbol{T}$.

In Fig. 2, we show experimental results that support Assumption 1. Specifically, the left figure shows the results when a 9-layer ReLU network is trained on the CIFAR100 dataset split into $N = 20$ tasks, while the right figure shows the results when a ResNet is trained on the ImageNet dataset split into $N = 50$ tasks. For each case, we focus on two models, $h_t$ and $h_{t'}$, where the task indices are set to $t = 1$ and $t' = N$. For each layer index $k$, we collect the weight matrices of both models: $\boldsymbol{W}_1^k$ and $\boldsymbol{W}_N^k$. Then, we train a linear transformation $\boldsymbol{T}$ to minimize the loss $\|\boldsymbol{W}_1^k - \boldsymbol{T}\boldsymbol{W}_N^k\|_2^2$ for 100 epochs and report the training loss curve, averaged across 50 random seeds. As for experimenting with the ResNet model, we generally apply the same procedure, but with an additional flattening procedure to address the difference in the shape of the weights of the convolutional layers; details are provided in the Appendix B.

While Assumption 1 may not universally apply to all architectures or scenarios, additional experiments with Vision Transformers (Dosovitskiy et al., 2020) show similar results (see Fig. 11 in the Appendix). Together, these findings

suggest that there exists $\boldsymbol{T}$ with $\boldsymbol{T}\boldsymbol{W}_{t'}^k \simeq \boldsymbol{W}_t^k$ in practice.

Now, we introduce two terms, the *layer cushion* and the *activation contraction*, which appear in our main theorem.

**Definition 5** (Layer Cushion (Arora et al., 2018)). *Let $h_t^k(\boldsymbol{x})$ denote the output of the $k$-th layer of the model $h_t(\boldsymbol{x})$ for some input $\boldsymbol{x} \in X_t$. The layer cushion of the model $h_t$ at layer $k$ is defined as*

$$\mu_{t,k} = \min\{\mu > 0 : \|\boldsymbol{W}_t^k\|_2 \|\phi(h_t^{k-1}(\boldsymbol{x}))\|_2$$
$$\leq \mu \|\boldsymbol{W}_t^k \phi(h_t^{k-1}(\boldsymbol{x}))\|_2 \; \forall \boldsymbol{x} \in X_t\}$$

*The layer cushion of the model $h_t$ is defined as*

$$\mu_t = \max_{k \in [L]} \mu_{t,k}.$$

**Definition 6** (Activation Contraction (Arora et al., 2018)). *Let $h_t^k(\boldsymbol{x})$ denote the output of the $k$-th layer of the model $h_t(\boldsymbol{x})$ for some input $\boldsymbol{x} \in X_t$. The activation contraction of model $h_t$ is defined as*

$$c_t = \min\{c > 0 : \|h_t^{k-1}(\boldsymbol{x})\|_2 \leq c \|\phi(h_t^{k-1}(\boldsymbol{x}))\|_2$$
$$\forall k \in [L], \forall \boldsymbol{x} \in X_t\}.$$

Using these definitions, we state our main theorem below:

**Theorem 1.** *Suppose Assumption 1 holds. Let $h_t$ be the model trained up to task $t$. Then, the representation discrepancy $D_t^k(h_t, \Delta t)$ defined in Def. 4 is bounded as*

$$D_t^k(h_t, \Delta t) \leq \mu_t c_t \|\mathcal{R}_t^k(h_t)\| \left( \frac{\omega_t^{k-1}(\Delta t)^2 + \omega_t^{k-1}(\Delta t)}{\omega_t^{k-1}(\Delta t)^2 + 1} \right),$$
(5)

*where* $\omega_t^{k-1}(\Delta t) = \frac{d(\mathcal{R}_t^{k-1}(h_t), \mathcal{R}_t^{k-1}(h_{t+\Delta t}))}{\|\mathcal{R}_t^{k-1}(h_t)\|}$. *Here,* $\|\mathcal{R}_t^k(h_t)\|, d(\mathcal{R}_t^k(h_t), \mathcal{R}_t^k(h_{t'})), \mu_t$ *and* $c_t$ *are given in Definitions 2, 3, 5 and 6, respectively.*

*Proof.* See Appendix A.1. $\square$

**Definition 7.** *The upper bound in the right-hand-side of Eqn. 5 is denoted by $U_t^k(\Delta t)$, i.e.,*

$$U_t^k(\Delta t) := \mu_t c_t \|\mathcal{R}_t^k(h_t)\| \left( \frac{\omega_t^{k-1}(\Delta t)^2 + \omega_t^{k-1}(\Delta t)}{\omega_t^{k-1}(\Delta t)^2 + 1} \right).$$

### 5.2. Evolution of Representation Discrepancy

The upper bound $U_t^k(\Delta t)$ on the representation discrepancy, given in Def. 7, can be interpreted as follows: For a fixed $t$ (the task index for which we measure the forgetting) and a fixed layer index $k$, the upper bound $U_t^k(\Delta t)$ becomes a function of $\Delta t$, which represents the number of additional tasks learned after learning task $t$.

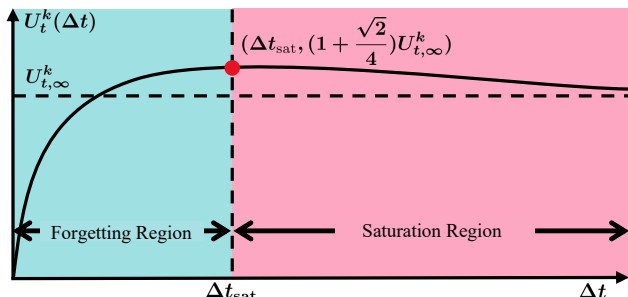

*Figure 3.* Illustration of Proposition 1. The graph of $U_t^k(\Delta t)$ (shown as black solid line) can be divided into two regions: (1) the *forgetting region* ($\Delta t < \Delta t_{\text{sat}}$), where $U_t^k(\Delta t)$ consistently increases as a function of $\Delta t$ and (2) the *saturation region* ($\Delta t \geq \Delta t_{\text{sat}}$), where $U_t^k(\Delta t)$ saturates to the asymptotic value $U_{t,\infty}^k$.

Note that $U_t^k(\Delta t)$ can be factored out into two terms: the first term $\mu_t c_t \|\mathcal{R}_t^k(h_t)\|$ that does not depend on $\Delta t$, and the second term $\frac{\omega_t^{k-1}(\Delta t)^2 + \omega_t^{k-1}(\Delta t)}{\omega_t^{k-1}(\Delta t)^2 + 1}$ that depends on $\Delta t$. To simplify the analysis of how $U_t^k(\Delta t)$ varies with $\Delta t$, we introduce the following assumption to model the relationship between $\omega_t^{k-1}(\Delta t)$ and $\Delta t$.

**Assumption 2.** *We assume that*

$$d(\mathcal{R}_t^{k-1}(h_t), \mathcal{R}_t^{k-1}(h_{t+\Delta t})) = \Theta(\Delta t) \quad (6)$$

In other words, we assume that the left-hand-side of Eq. (6) is both upper and lower bounded by a linear function of $\Delta t$. The upper bound has already been established to be linear with $\Delta t$ according to Theorem 4.1 of (Guha & Lakshman, 2024), while the linearity of the lower bound has not yet been established in the literature.

Based on the above results, the following proposition explains how $U_t^k(\Delta t)$ evolves over $\Delta t$.

**Proposition 1.** *Suppose Assumption 2 holds. The upper bound $U_t^k(\Delta t)$ of representation discrepancy monotonically increases with respect to $\Delta t$, reaching a peak value of $(1 + \frac{\sqrt{2}}{4})\mu_t c_t \|\mathcal{R}_t^k(h_t)\|$ before gradually saturating to $\mu_t c_t \|\mathcal{R}_t^k(h_t)\|$.*

*Proof.* Note that according to Assumption 2, $d(\mathcal{R}_t^{k-1}(h_t), \mathcal{R}_t^{k-1}(h_{t+\Delta t}))$ increases linearly with $\Delta t$. This implies that $\omega_t^{k-1}(\Delta t) = \frac{d(\mathcal{R}_t^{k-1}(h_t), \mathcal{R}_t^{k-1}(h_{t+\Delta t}))}{\|\mathcal{R}_t^{k-1}(h_t)\|}$ also increases with $\Delta t$. Since $\frac{U_t^k(\Delta t)}{\mu_t c_t \|\mathcal{R}_t^k(h_t)\|} = f \circ \omega_t^{k-1}(\Delta t)$ for $f(x) = \frac{x^2 + x}{x^2 + 1} = 1 + \frac{x-1}{x^2+1}$, the shape of $U_t^k$ solely depends on the shape of $f$, which concludes the proof. $\square$

Fig. 3 shows the implications of Proposition 1: the upper bound $U_t^k$ on the representation discrepancy exhibits two different phases. The first phase, which we call the *forgetting phase*, is where $U_t^k(\Delta t)$ monotonically increases as a

function of $\Delta t$. The second phase, which we call the *saturation phase*, is where $U_t^k$ slowly saturates to an asymptotic value.

Therefore, we focus on two quantities that capture the evolution of the upper bound $U_t^k$ of the representation discrepancy. The first quantity is the *asymptotic representation discrepancy*

$$U_{t,\infty}^k := \lim_{\Delta t \to \infty} U_t^k(\Delta t) = \mu_t c_t \|\mathcal{R}_t^k(h_t)\|, \quad (7)$$

defined as $U_t^k(\Delta t)$ in the asymptotic regime of large $\Delta t$. The second quantity is the *convergence rate* of the representation discrepancy, which is defined as how fast $U_t^k(\Delta t)$ enters the saturation region, denoted by

$$r_t^k := \frac{1}{\Delta t_{\text{sat}}}, \quad (8)$$

where

$$\Delta t_{\text{sat}} := \arg\max_{\Delta t > 0} U_t^k(\Delta t) \quad (9)$$

is the task index when the model enters the saturation region, as shown in Fig. 3.

Now we investigate how the asymptotic representation discrepancy $U_{t,\infty}^k$ and the convergence rate $r_t^k$ behave as the layer index $k$ and the width $m$ of the neural network vary.

First, we state how $U_{t,\infty}^k$ varies as a function of $k$.

**Corollary 1.** *Let the task index $t$ and layer index $k$ be fixed. Then, the $k$-dependency of the asymptotic representation discrepancy $U_{t,\infty}^k$ is fully captured by $\|\mathcal{R}_t^k(h_t)\|$ defined in Def. 2, the maximum norm of the activation at layer $k$. In addition, $U_{t,\infty}^k$ is linearly proportional to $\|\mathcal{R}_t^k(h_t)\|$, i.e., $U_{t,\infty}^k \propto \|\mathcal{R}_t^k(h_t)\|$.*

Note that our empirical results (plotted in Fig. 5 in Sec. 6) show that $\|\mathcal{R}_t^k(h_t)\|$ is linearly proportional to $k$. Combining this with the above corollary implies that the asymptotic representation forgetting $U_{t,\infty}^k$ is a linear function of $k$, i.e., the model forgets more in the deeper layers.

Second, we analyze how the convergence rate $r_t^k$ of the representation discrepancy behaves as the layer index $k$ and the width $m$ varies. We begin with an assumption that was empirically justified in Guha & Lakshman (2024).

**Assumption 3** (Assumption 4.3 in Guha & Lakshman (2024)). *Let $h(\boldsymbol{x}) = \boldsymbol{W}^L \phi(\boldsymbol{W}^{L-1} \cdots \phi(\boldsymbol{W}^1 \boldsymbol{x}))$ be an $L$-layer ReLU network with each hidden layer $k$ having the same width $m$, and let $t \in [N]$ be any task index. Consider the continual learning setup, where the weights are updated via gradient-based learning methods. Then, the weight matrices ($\boldsymbol{W}_t^k$ and $\boldsymbol{W}_{t+1}^k$) of the network $h$ before/after training on task $t + 1$ satisfies the following for all layers $k$:*

$$\frac{\|\boldsymbol{W}_{t+1}^k - \boldsymbol{W}_t^k\|_F}{\|\boldsymbol{W}_t^k\|_2} \leq \gamma m^{-\beta}, \quad (10)$$

*where $\gamma, \beta \in \mathbb{R}$ are positive constants[2].*

The above assumption implies that for each layer $k$, the normalized difference between the weight matrix $\boldsymbol{W}_{t+1}$ *after* learning task $t + 1$ and the weight matrix $\boldsymbol{W}_t$ *before* learning task $t + 1$ is upper bounded by a quantity that decreases as the model width $m$ grows. This assumption allows us to formulate how the weight matrices of each layer change after learning the subsequent tasks. Under this assumption, we derive the upper bound on the convergence rate $r_t^k$ of the representation discrepancy.

**Theorem 2.** *Suppose Assumption 3 holds. Let $\lambda_{t,t'}^k = \frac{\|\boldsymbol{W}_{t'}^k\|_2}{\|\boldsymbol{W}_t^k\|_2}$ denote the ratio of the spectral norms of the $k$-th layer weight matrices for task $t$ and $t'$, and let $\lambda_t = \max_{i \in [L], t' \in [N]} \lambda_{t,t'}^i$. Then, the convergence rate $r_t^k$ of the representation discrepancy is bounded as*

$$r_t^k \leq (\sqrt{2} - 1)\gamma m^{-\beta} \sum_{i=1}^{k} \frac{(\lambda_t \mu_t c_t)^i}{\lambda_t}, \quad (11)$$

*where $\gamma, \beta$ are positive constants defined in Assumption 3, and $\mu_t, c_t$ are constants defined in Definitions 5 and 6.*

*Proof.* See Appendix A.2. □

Since the constants $\lambda_t, \mu_t, c_t$ are positive for any $t$, the upper bound on the right-hand side of Eqn. 11 increases with the layer index $k$. This implies that the layer closer to the input (*i.e.*, lower $k$) forgets at a slower pace, compared with the layer closer to the output (*i.e.*, higher $k$). Additionally, the exponent of the width $m$ on the right-hand side of Eqn. 11 is negative, indicating that the model with larger width forgets the representation more slowly.

## 6. Experiments

In this section, we present experimental results that support our theoretical findings. We measured both the representation forgetting and the upper bound of the representation discrepancy under the same continual learning setup. Due to limited spacing, we report results on representation forgetting here and defer those on representation discrepancy to Appendix C. In Sec. 6.1, we empirically observe the representation forgetting of the first task during the continual learning process and check whether it coincides with the behavior deduced from our analysis, as demonstrated in Proposition 1. In Sec. 6.2, we provide empirical results that show the linear relationship between the overall amount of representation forgetting and the size of the representation space, which supports our theoretical findings in Corollary 1. In Sec. 6.3, we investigate whether the layer index $k$ and

---

[2]For more details on the actual values of $\beta$ and $\gamma$, refer to Guha & Lakshman (2024).

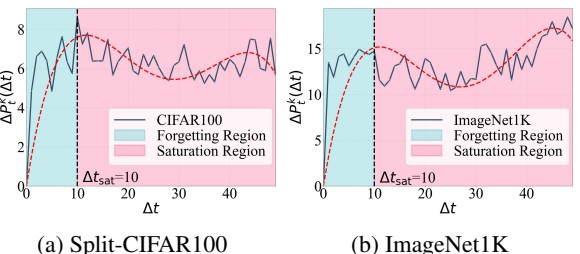

(a) Split-CIFAR100       (b) ImageNet1K

*Figure 4.* The evolution of representation forgetting $\Delta P_t^k(\Delta t)$ measured on Split-CIFAR100 and ImageNet1K datasets. We measure $\Delta P_t^k(\Delta t)$ for $t = 1$ and $k = L$. The solid line shows $\Delta P_t^k(\Delta t)$, while the red dashed line shows the 4-th degree polynomial that best fits to the solid line, to check the overall tendency of the representation forgetting. Similar to the plot in Fig. 3, we divide the graph into two regions based on the point where the first local maximum of the 4-th degree polynomial has occurred. For both datasets, we see that this is when $\Delta t_{sat} = 10$, as indicated by the black dashed line. The shape of the plot for $\Delta P_t^k$ is similar to the shape of the upper bound $U_t^k$ of the representation discrepancy we analyzed in Proposition 1.

the model width $m$ satisfies the monotonic relationships (in accordance to Thm. 2) to the convergence rate of the representation forgetting. Before we show our results, we give an outline of the setup for our continual learning experiments.

**Implementation details** We test on Split-CIFAR100 dataset (Ramasesh et al., 2020) and a downsampled version of the original ImageNet1K dataset (Chrabaszcz et al., 2017). For each dataset, the classes are partitioned into $N = 50$ categories, with each category containing 2 classes for Split-CIFAR100 and 5 classes for ImageNet1K, following their original indices. Our model is based on the ResNet architecture (He et al., 2016), modified to maintain consistent latent feature dimensions throughout the network by adjusting parameters such as stride and kernel size; see detailed structure in Fig. 7 in Appendix. Specifically, we define the network width $m$ as *(# channels)* $\times 32 \times 32$, and the number of layers $L$ as the number of blocks. By default, we set *(# channels)* $= 8$ and use $L = 9$ blocks. After pre-training ResNet through the continual learning process, we measure the performance of each hidden layer feature using linear probing; we extract the features at $k$-th block of ResNet, and train a linear classifier for task $t = 1$ on top of the extracted features.

### 6.1. Evolution of Representation Forgetting

In Fig. 4, we plot the evolution of the representation forgetting $\Delta P_t^k(\Delta t)$ as $\Delta t$ grows, when $t = 1$ and $k = 9$, tested on Split-CIFAR100 and ImageNet1K datasets, respectively. We see that the forgetting curves for both experiments exhibit similar properties: initially, they both increase with $\Delta t$, but after some threshold value, *i.e.,* $\Delta t = 10$, the curves

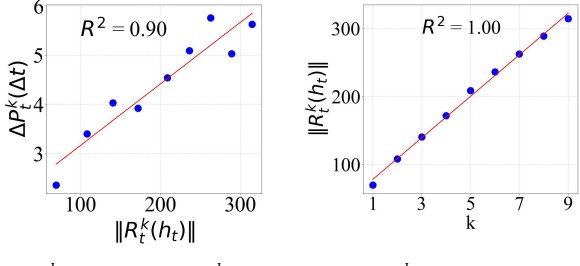

(a) $\Delta P_t^k(\Delta t)$ versus $\|\mathcal{R}_t^k(h_t)\|$    (b) $\|\mathcal{R}_t^k(h_t)\|$ versus $k$

*Figure 5.* The relationship between the amount of representation forgetting $\Delta P_t^k(\Delta t)$, the size of the representation space $\|\mathcal{R}_t^k(h_t)\|$, and the layer index $k$. We set $t = 1$, $\Delta t = N - 1$, and evaluated each quantity for $k = 1, \cdots, 9$. The left plot shows a strong linear relationship between $\Delta P_t^k(\Delta t)$ and $\|\mathcal{R}_t^k(h_t)\|$, which coincides with Corollary 1. The right plot also shows a strong linear relationship between $\|\mathcal{R}_t^k(h_t)\|$ and $k$. Similar results for ImageNet1K datasets are given in Fig. 12 in Appendix.

stay within a bounded region[3]. This behavior is consistent with our theoretical results on the upper bound $U_t^k$ of the representation discrepancy, as described in Proposition 1.

### 6.2. Bigger Spaces Experience Greater Forgetting

In Fig. 5a, we provide experimental results that relate the size $\|\mathcal{R}_t^k(h_t)\|$ of the representation space and the amount of representation forgetting $\Delta P_t^k(\Delta t)$ at the final phase of the continual learning, *i.e.,* when $\Delta t = N - 1$. Specifically, for each layer $k = 1, 2, \cdots, 9$, we calculate the difference in the linear probing accuracy of task $t = 1$, before and after learning the remaining $N - 1 = 49$ tasks. We then plot this against the norm of the largest feature in $\mathcal{R}_t^k(h_t)$ and assess the linearity by fitting a line using linear regression. As shown in Fig. 5a, the high $R^2$ value confirms the strong linear relationship between $\Delta P_1^k(\Delta t)$ and $\|\mathcal{R}_1^k(h_1)\|$. This coincides with our findings in Corollary 1 for $U_{t,\infty}^k$, an upper bound on a proxy of representation forgetting $\Delta P_t^k$.

In Fig. 5b, we plot $\|\mathcal{R}_t^k(h_t)\|$ against the layer index $k$. Surprisingly, we observe a strong linear relationship between them. Combining this result with Corollary 1 implies that the amount of asymptotic representation forgetting $U_{t,\infty}^k$ linearly increases with the layer index $k$.

### 6.3. Effect of Network Size on the Convergence Rate

We empirically show how the convergence rate of the representation forgetting $\Delta P_t^k$ changes as the layer index $k$ and the network width $m$ varies, for Split-CIFAR100 dataset. We first draw $\Delta P_t^k$ as well as the 4-th degree polynomial

---

[3]Empirical results indicate that this steady-state behavior occurs while the model retains a non-trivial amount of information from task 1. See Fig. 10 in Appendix.

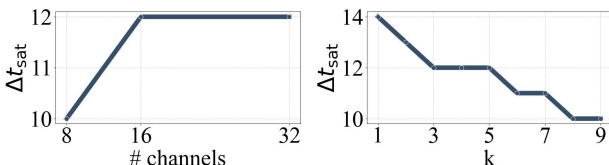

*Figure 6.* The impact of # channels (**Left**) and $k$ (**Right**) on $\Delta t_{\text{sat}}$. For each $k$ and # channels, we report $\Delta t$ when the best fitted 4-th degree polynomial achieves its first local maximum, similar to the point indicated by the black dashed line in Fig. 4.

that best fits $\Delta P_t^k$, similar to the plot shown in Fig. 4a. Then, we get $\Delta t_{\text{sat}}$ defined in Eqn. 9, which is estimated by the local maximum of the 4-th degree polynomial that best fits $\Delta P_t^k$. Fig. 6 shows $\Delta t_{\text{sat}}$ for various width $m$ (or equivalently, *# channels* in ResNet) and layer index $k$. Our results indicate that $\Delta t_{\text{sat}}$ decreases with the layer index $k$, indicating that the convergence rate $r_t^k = \frac{1}{\Delta t_{\text{sat}}}$ increases with $k$. In addition, increasing *# channels* results in longer $\Delta t_{\text{sat}}$, indicating that increasing the width decreases the convergence rate. These results support our theoretical findings from Thm. 2. Altogether, our results indicate that the forgetting occurs *more rapidly* to a *higher* degree as the layer index $k$ increases, while increasing the width $m$ of the network slows down the forgetting process.

## 7. Conclusion

In this paper, we present the first theoretical analysis of representation forgetting in continual learning. By introducing *representation discrepancy*, a metric measuring dissimilarity between representation spaces under optimal linear transformation, we characterize the evolution of forgetting as the model learns new tasks. Our analysis shows that the forgetting curve exhibits two distinct phases (forgetting and saturation) and demonstrates that forgetting intensifies in deeper layers while larger network width delays forgetting. These theoretical findings are validated through experiments on image datasets.

## Acknowledgments

This work was supported by the National Research Foundation of Korea (NRF) grant funded by the Korea government (MSIT) (RS-2024-00345351, RS-2024-00408003), the MSIT (Ministry of Science and ICT), Korea, under the ICAN (ICT Challenge and Advanced Network of HRD) support program (RS-2023-00259934, RS-2025-02283048) supervised by the IITP (Institute for Information & Communications Technology Planning & Evaluation), the Yonsei University Research Fund (2025-22-0025).

## Impact Statement

This paper presents theoretical understanding of the representation forgetting in continual learning scenarios. Our results reveal the two-phase structure of the representation forgetting as well as the behavior of the forgetting during such phases. This discovery provides a deeper understanding of how models lose previously learned knowledge over time and sheds light on the mechanisms behind catastrophic forgetting. It may have implications on designing safer continual learning algorithms, particularly in scenarios where retaining sensitive information is critical. By addressing the challenges associated with knowledge retention, this research contributes to advancing the reliability and robustness of lifelong learning systems.

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

# A. Proofs

## A.1. Proof of Theorem 1

The proof of Thm.1 can be split into 3 parts: (1) derive an upper bound of $D_t^k(h_t, \Delta t)$ as an optimization problem over the linear transformation $\boldsymbol{T}$, (2) utilize Lemma 2 to derive a suboptimal solution for the optimization problem (3) apply compression approaches introduced by Arora et al. (2018) to remove the dependency on the weight norm of the bound.

### A.1.1. DERIVING AN UPPER BOUND IN TERMS OF $\boldsymbol{T}$

**Lemma 1.** *Let $h(\boldsymbol{x}) = \boldsymbol{W}^L \phi(\boldsymbol{W}^{L-1} \cdots \phi(\boldsymbol{W}^1 \boldsymbol{x}))$ be a randomly initialized L-layer ReLU Network with each hidden layer $k$ having the same width $m$. Suppose $h(\boldsymbol{x})$ is trained sequentially on the datasets $\mathcal{D}_1, \cdots, \mathcal{D}_N$ and $h_t$ represents the model trained up to $\mathcal{D}_t$. Define $\|\mathcal{R}_t^k(h_t)\| := \max\{\|h_t^k(\boldsymbol{x})\|_2 : \boldsymbol{x} \in X_t\}$, as the norm of the biggest feature in $\mathcal{R}_t^k(h_t)$, and $d(\mathcal{R}_t^L(h_t), \mathcal{R}_t^L(h_{t'})) := \max\{\|\boldsymbol{z} - \boldsymbol{z}'\|_2 : \boldsymbol{z} \in \mathcal{R}_t^L(h_t), \boldsymbol{z}' \in \mathcal{R}_{t'}^L(h_{t'}^k)\}$, as the distance between the two most distant features sampled at each space. Then, the $k$-th layer representation discrepancy of task $t$ in model $h_t$ when trained on additional $\Delta t$ tasks is bounded as*

$$D_t^k(h_t, \Delta t) \leq \min_{\boldsymbol{T}} d(\mathcal{R}_t^{k-1}(h_t), \mathcal{R}_t^{k-1}(h_{t+\Delta t})) \, \|\boldsymbol{T} \boldsymbol{W}_{t'}^k\|_2 + \|\mathcal{R}_t^{k-1}(h_t)\| \, \|\boldsymbol{T} \boldsymbol{W}_{t'}^k - \boldsymbol{W}_t^k\|_2 \tag{12}$$

*where $\boldsymbol{T} \in \mathbb{R}^{w_k \times w_k}$ is any linear transformation from the $k$-th representation space to itself*

*Proof.*

$$\begin{aligned}
D_t^k(h_t, \Delta t) &= \min_{\boldsymbol{T}} \max_{\boldsymbol{x} \in X_t} \|\boldsymbol{T} \circ h_{t+\Delta t}^k(\boldsymbol{x}) - h_t^k(\boldsymbol{x})\|_2 \\
&= \min_{\boldsymbol{T}} \max_{\boldsymbol{x} \in X_t} \|\boldsymbol{T} \boldsymbol{W}_{t+\Delta t}^k \phi(h_{t+\Delta t}^{k-1}(\boldsymbol{x})) - \boldsymbol{W}_t^k \phi(h_t^{k-1}(\boldsymbol{x}))\|_2 \\
&= \min_{\boldsymbol{T}} \max_{\boldsymbol{x} \in X_t} \|\boldsymbol{T} \boldsymbol{W}_{t+\Delta t}^k \phi(h_{t+\Delta t}^{k-1}(\boldsymbol{x})) - \boldsymbol{T} \boldsymbol{W}_{t+\Delta t}^k \phi(h_t^{k-1}(\boldsymbol{x})) + \boldsymbol{T} \boldsymbol{W}_{t+\Delta t}^k \phi(h_t^{k-1}(\boldsymbol{x})) - \boldsymbol{W}_t^k \phi(h_t^{k-1}(\boldsymbol{x}))\|_2 \\
&\leq \min_{\boldsymbol{T}} \max_{\boldsymbol{x} \in X_t} \|\boldsymbol{T} \boldsymbol{W}_{t+\Delta t}^k \phi(h_{t+\Delta t}^{k-1}(\boldsymbol{x})) - \boldsymbol{T} \boldsymbol{W}_{t+\Delta t}^k \phi(h_t^{k-1}(\boldsymbol{x}))\|_2 + \|\boldsymbol{T} \boldsymbol{W}_{t+\Delta t}^k \phi(h_t^{k-1}(\boldsymbol{x})) - \boldsymbol{W}_t^k \phi(h_t^{k-1}(\boldsymbol{x}))\|_2 \\
&\leq \min_{\boldsymbol{T}} \max_{\boldsymbol{x} \in X_t} \|\boldsymbol{T} \boldsymbol{W}_{t+\Delta t}^k\|_2 \|\phi(h_{t+\Delta t}^{k-1}(\boldsymbol{x})) - \phi(h_t^{k-1}(\boldsymbol{x}))\|_2 + \|\boldsymbol{T} \boldsymbol{W}_{t+\Delta t}^k - \boldsymbol{W}_t^k\|_2 \|\phi(h_t^{k-1}(\boldsymbol{x}))\|_2 \tag{13} \\
&\leq \min_{\boldsymbol{T}} \max_{\boldsymbol{x} \in X_t} \|\boldsymbol{T} \boldsymbol{W}_{t+\Delta t}^k\|_2 \|h_{t+\Delta t}^{k-1}(\boldsymbol{x}) - h_t^{k-1}(\boldsymbol{x})\|_2 + \|\boldsymbol{T} \boldsymbol{W}_{t+\Delta t}^k - \boldsymbol{W}_t^k\|_2 \|h_t^{k-1}(\boldsymbol{x})\|_2 \tag{14} \\
&\leq \min_{\boldsymbol{T}} d(\mathcal{R}_t^{k-1}(h_t), \mathcal{R}_t^{k-1}(h_{t+\Delta t})) \|\boldsymbol{T} \boldsymbol{W}_{t+\Delta t}^k\|_2 + \|\mathcal{R}_t^{k-1}(h_t)\| \, \|\boldsymbol{T} \boldsymbol{W}_{t+\Delta t}^k - \boldsymbol{W}_t^k\|_2
\end{aligned}$$

We used the sub-multiplicative property of the matrix 2-norm to derive Equation (12), and 1-Lipschitz property of $\phi$ to derive Equation (14). $\square$

### A.1.2. FINDING A SUBOPTIMAL SOLUTION

We use the following Lemma 2 to derive a suboptimal solution for Eqn. 12.

**Lemma 2.** *Let $\boldsymbol{A} \in \mathbb{R}^{n \times n}$ be an arbitrary matrix, and $c_1, c_2 \in \mathbb{R}_{>0}$ be any positive constants. Then,*

$$\min_{\boldsymbol{X} \in \mathbb{R}^{n \times n}} c_1 \|\boldsymbol{X}\|_2 + c_2 \|\boldsymbol{X} - \boldsymbol{A}\|_2 \leq c_2 \|\boldsymbol{A}\|_2 \left( \frac{\omega^2 + \omega}{\omega^2 + 1} \right) \tag{15}$$

*where $\omega = \frac{c_1}{c_2}$*

*Proof.* Since the $l_2$ norm for a matrix is both non-differentiable and non-convex, we derive an optimal solution to its close convex-differentiable form,

$$\min_{\boldsymbol{X} \in \mathbb{R}^{m \times n}} c_1 \|\boldsymbol{X}\|_2 + c_2 \|\boldsymbol{X} - \boldsymbol{T}\|_2 \leq c_1 \|\boldsymbol{X}^*\|_2 + c_2 \|\boldsymbol{X}^* - \boldsymbol{T}\|_2$$

where $\boldsymbol{X}^* = \underset{\boldsymbol{X} \in \mathbb{R}^{m \times n}}{\arg\min} c_1^2 \|\boldsymbol{X}\|_F^2 + c_2^2 \|\boldsymbol{X} - \boldsymbol{A}\|_F^2$

Now, since $f(\boldsymbol{X}) = c_1^2 \|\boldsymbol{X}\|_F^2 + c_2^2 \|\boldsymbol{X} - \boldsymbol{A}\|_F^2$ is convex and differentiable, it is enough to find $\boldsymbol{X}^*$ such that $\nabla f(\boldsymbol{X}^*) = 0$

We claim that $\boldsymbol{X}^* = \frac{c_2^2}{c_1^2 + c_2^2}\boldsymbol{A}$ satisfies the above condition.

$$\begin{aligned}
\nabla f(\boldsymbol{X}^*) &= 2c_1^2\boldsymbol{X}^* + 2c_2^2(\boldsymbol{X}^* - \boldsymbol{A}) \\
&= \Big(\frac{2c_1^2 c_2^2}{c_1^2 + c_2^2} - \frac{2c_1^2 c_2^2}{c_1^2 + c_2^2}\Big)\boldsymbol{A} \\
&= 0
\end{aligned}$$

Hence, we can now derive the upper bound as follows,

$$\begin{aligned}
\min_{\boldsymbol{X} \in \mathbb{R}^{m \times n}} c_1\|\boldsymbol{X}\|_2 + c_2\|\boldsymbol{X} - \boldsymbol{T}\|_2 &\leq c_1\|\boldsymbol{X}^*\|_2 + c_2\|\boldsymbol{X}^* - \boldsymbol{T}\|_2 \\
&= \frac{c_1 c_2^2 + c_2 c_1^2}{c_1^2 + c_2^2}\|\boldsymbol{A}\|_2 \\
&= c_2\frac{\lambda^2 + \lambda}{\lambda^2 + 1}\|\boldsymbol{A}\|_2
\end{aligned}$$

where $\lambda = \frac{c_1}{c_2}$ $\qquad\qquad\qquad\qquad\square$

Substituting $\boldsymbol{A} = \boldsymbol{W}_{t+\Delta t}^k, \boldsymbol{X} = \boldsymbol{T}\boldsymbol{W}_t^k, c_1 = d(\mathcal{R}_t^{k-1}(h_t), \mathcal{R}_t^{k-1}(h_{t+\Delta t})), c_2 = \|\mathcal{R}_t^{k-1}(h_t)\|$ in Lemma 2, we get the following Corollary 2.

**Corollary 2.** *Let $h(\boldsymbol{x}) = \boldsymbol{W}^L\phi(\boldsymbol{W}^{L-1}\cdots\phi(\boldsymbol{W}^1\boldsymbol{x}))$ be a randomly initialized L-layer ReLU Network with each hidden layer $k$ having the same width $m$. Suppose Assumption 1 holds. Then, the $k$-th layer representation discrepancy of task $t$ in model $h_t$ when trained on additional $\Delta t$ tasks is bounded as*

$$D_t^k(h_t, \Delta t) \leq \|\mathcal{R}_t^{k-1}(h_t)\|\|\boldsymbol{W}_t^k\|_2\Big(\frac{\omega_t^{k-1}(\Delta t)^2 + \omega_t^{k-1}(\Delta t)}{\omega_t^{k-1}(\Delta t)^2 + 1}\Big), \qquad (16)$$

*where $\omega_t^{k-1}(\Delta t) = \frac{d(\mathcal{R}_t^{k-1}(h_t), \mathcal{R}_t^{k-1}(h_{t+\Delta t}))}{\|\mathcal{R}_t^{k-1}(h_t)\|}$*

### A.1.3. REMOVING THE DEPENDENCY ON THE WEIGHT NORM

For the final step, we use Proposition 2 to remove the dependency on the weight norm in Eqn. 16.

**Proposition 2.** *Following the setting of Corollary 2, we have,*

$$\max_{\boldsymbol{x} \in X_t} \|h_t^{k-1}(\boldsymbol{x})\|_2 \leq \frac{\mu_t c_t\|\mathcal{R}_t^k(h_t)\|}{\|\boldsymbol{W}_t^k\|_2} \qquad (17)$$

*where $\mu_t, c_t$ are constants depending on the dataset $\mathcal{D}_t$ and the model $h_t$*

*Proof.* The proof follows trivially from the definitions of the layer cushion Def. 5 and the activation contraction Def. 6. Namely,

$$\begin{aligned}
\max_{\boldsymbol{x} \in X_t} \|h_t^{k-1}(\boldsymbol{x})\|_2 &= \frac{1}{\|\boldsymbol{W}_t^k\|_2}\max_{\boldsymbol{x} \in X_t}\|\boldsymbol{W}_t^k\|_2\|h_t^{k-1}(\boldsymbol{x})\|_2 \\
&\leq \frac{1}{\|\boldsymbol{W}_t^k\|_2}\max_{\boldsymbol{x} \in X_t} c_t\|\boldsymbol{W}_t^k\|_2\|\phi(h_t^{k-1}(\boldsymbol{x}))\|_2 \\
&\leq \frac{1}{\|\boldsymbol{W}_t^k\|_2}\max_{\boldsymbol{x} \in X_t} \mu_t c_t\|\boldsymbol{W}_t^k\phi(h_t^{k-1}(\boldsymbol{x}))\|_2 \\
&\leq \frac{\mu_t c_t\|\mathcal{R}_t^k(h_t)\|}{\|\boldsymbol{W}_t^k\|_2}
\end{aligned}$$

$\qquad\qquad\qquad\qquad\qquad\qquad\qquad\qquad\qquad\qquad\qquad\qquad\qquad\qquad\qquad\qquad\square$

Proposition 2 trivially follows from Def. 5 and Def. 6. Using Proposition 2 to replace $\|\mathcal{R}_t^{k-1}(h_t)\|$ with $\frac{\mu_t c_t \|\mathcal{R}_t^k(h_t)\|}{\|\boldsymbol{W}_t^k\|_2}$ in Lemma 2, we get Thm. 1.

**Theorem 3.** *Suppose Assumption 1 holds. Let $h_t$ be the model trained up to $\mathcal{D}_t$. Define $\|\mathcal{R}_t^k(h_t)\| := \max\{\|h_t^k(\boldsymbol{x})\|_2 : \boldsymbol{x} \in X_t\}$, as the norm of the biggest feature in $\mathcal{R}_t^k(h_t)$, and $d(\mathcal{R}_t^L(h_t), \mathcal{R}_t^L(h_{t'})) := \max\{\|\boldsymbol{z}_1 - \boldsymbol{z}_2\|_2 : \boldsymbol{z}_1 \in \mathcal{R}_t^k(h_{t_1}), \boldsymbol{z}_2 \in \mathcal{R}_t^k(h_{t_2})\}$, as the distance between the two most distant features sampled at each space. Then, the $k$-th layer representation discrepancy of task $t$ in model $h_t$ when trained on additional $\Delta t$ tasks is bounded as*

$$D_t^k(h_t, \Delta t) \leq \mu_t c_t \|\mathcal{R}_t^k(h_t)\| \left( \frac{\omega_t^{k-1}(\Delta t)^2 + \omega_t^{k-1}(\Delta t)}{\omega_t^{k-1}(\Delta t)^2 + 1} \right), \tag{18}$$

*where $\omega_t^{k-1}(\Delta t) = \frac{d(\mathcal{R}_t^{k-1}(h_t), \mathcal{R}_t^{k-1}(h_{t+\Delta t}))}{\|\mathcal{R}_t^{k-1}(h_t)\|}$, and $\mu_t, c_t$ are constants depending on the dataset $\mathcal{D}_t$ and the model $h_t$, as defined in Def. 5 and 6.*

### A.2. Proof of Theorem 2

The proof of Thm. 2 consists of two steps. First, in Thm. 4, we derive an upper bound of $\omega_t^k(\Delta t)$ through perturbation analysis, using techniques similar to Guha & Lakshman (2024). Then, we derive the upper bound of $r_t^k$ by using the fact that $\Delta t_{\text{sat}} := \arg\max_{\Delta t > 0} U_t^k(\Delta t)$ holds when $\omega_t^k(\Delta t_{\text{sat}}) = \sqrt{2} + 1$.

**Theorem 4.** *Let $\lambda_{t,t'}^k = \frac{\|\boldsymbol{W}_{t'}^k\|_2}{\|\boldsymbol{W}_t^k\|_2}$ denote the ratio of the spectral norms of the $k$-th layer weight matrices for task indexes $t$ and $t'$, and $\lambda_t = \max_{i \in [L], t' \in [N]} \lambda_{t,t'}^i$. Following the setting of Thm. 1, $\omega_t^k(\Delta t) = \frac{d(\mathcal{R}_t^k(h_t), \mathcal{R}_t^k(h_{t+\Delta t}))}{\|\mathcal{R}_t^k(h_t)\|}$ has an upper bound as*

$$\omega_t^k(\Delta t) \leq \gamma m^{-\beta} \Delta t \sum_{i=1}^k \frac{(\lambda_t \mu_t c_t)^i}{\lambda_t}, \tag{19}$$

*where $\gamma, \beta$ are positive constants defined in Assumption 3, and $\mu_t, c_t$ are constants defined in Definitions 5 and 6, which depend on the dataset $\mathcal{D}_t$ and the model $h_t$.*

*Proof.* From the definition of $\omega_t^k(\Delta t)$, we can see that,

$$\omega_t^k(\Delta t) = \frac{\max_{\boldsymbol{x} \in X_t} \|h_t^k(\boldsymbol{x}) - h_{t+\Delta t}^k(\boldsymbol{x})\|_2}{\max_{\boldsymbol{x} \in X_t} \|h_t^k(\boldsymbol{x})\|_2}$$

$$\leq \max_{\boldsymbol{x} \in X_t} \frac{\|h_t^k(\boldsymbol{x}) - h_{t+\Delta t}^k(\boldsymbol{x})\|_2}{\|h_t^k(\boldsymbol{x})\|_2}$$

Now, we prove that,

$$\max_{\boldsymbol{x} \in X_t} \frac{\|h_t^k(\boldsymbol{x}) - h_{t+\Delta t}^k(\boldsymbol{x})\|_2}{\|h_t^k(\boldsymbol{x})\|_2} \leq \left( \frac{\gamma}{\lambda_t} m^{-\beta} \Delta t \right) \sum_{i=1}^k (\lambda_t \mu_t c_t)^i$$

This can be seen from the following.

$$\frac{\|h_t^k(\boldsymbol{x}) - h_{t+\Delta t}^k(\boldsymbol{x})\|_2}{\|h_t^k(\boldsymbol{x})\|_2} \leq \frac{\|h_t^k(\boldsymbol{x}) - h_{t+\Delta t}^k(\boldsymbol{x})\|_2}{\frac{1}{\mu_t c_t}\|\boldsymbol{W}_t^k\|_2\|h_t^{k-1}(\boldsymbol{x})\|_2}$$

$$= \mu_t c_t \frac{\|\boldsymbol{W}_t^k\phi(h_t^{k-1}(\boldsymbol{x})) - \boldsymbol{W}_{t+\Delta t}^k\phi(h_{t+\Delta t}^k(\boldsymbol{x}))\|_2}{\|\boldsymbol{W}_t^k\|_2\|h_t^{k-1}(\boldsymbol{x})\|_2}$$

$$= \mu_t c_t \frac{\|\boldsymbol{W}_t^k\phi(h_t^{k-1}(\boldsymbol{x})) - \boldsymbol{W}_{t+\Delta t}^k\phi(h_t^{k-1}(\boldsymbol{x})) + \boldsymbol{W}_{t+\Delta t}^k\phi(h_t^{k-1}(\boldsymbol{x})) - \boldsymbol{W}_{t+\Delta t}^k\phi(h_{t+\Delta t}^{k-1}(\boldsymbol{x}))\|_2}{\|\boldsymbol{W}_t^k\|_2\|h_t^{k-1}(\boldsymbol{x})\|_2}$$

$$\leq \mu_t c_t \frac{\|\boldsymbol{W}_t^k\phi(h_t^{k-1}(\boldsymbol{x})) - \boldsymbol{W}_{t+\Delta t}^k\phi(h_t^{k-1}(\boldsymbol{x}))\|_2 + \|\boldsymbol{W}_{t+\Delta t}^k\phi(h_t^{k-1}(\boldsymbol{x})) - \boldsymbol{W}_{t+\Delta t}^k\phi(h_{t+\Delta t}^{k-1}(\boldsymbol{x}))\|_2}{\|\boldsymbol{W}_t^k\|_2\|h_t^{k-1}(\boldsymbol{x})\|_2}$$

$$\leq \mu_t c_t \frac{\|\boldsymbol{W}_t^k - \boldsymbol{W}_{t+\Delta t}^k\|_2\|\phi(h_t^{k-1}(\boldsymbol{x}))\|_2 + \|\boldsymbol{W}_{t+\Delta t}^k\|_2\|\phi(h_t^{k-1}(\boldsymbol{x})) - \phi(h_{t+\Delta t}^{k-1}(\boldsymbol{x}))\|_2}{\|\boldsymbol{W}_t^k\|_2\|h_t^{k-1}(\boldsymbol{x})\|_2} \tag{20}$$

$$\leq \mu_t c_t \frac{\|\boldsymbol{W}_t^k - \boldsymbol{W}_{t+\Delta t}^k\|_2\|h_t^{k-1}(\boldsymbol{x})\|_2 + \|\boldsymbol{W}_{t+\Delta t}^k\|_2\|h_t^{k-1}(\boldsymbol{x}) - h_{t+\Delta t}^{k-1}(\boldsymbol{x})\|_2}{\|\boldsymbol{W}_t^k\|_2\|h_t^{k-1}(\boldsymbol{x})\|_2} \tag{21}$$

$$\leq \mu_t c_t \left(\frac{\|\boldsymbol{W}_t^k - \boldsymbol{W}_{t+\Delta t}^k\|_2}{\|\boldsymbol{W}_t^k\|_2} + \frac{\|\boldsymbol{W}_{t+\Delta t}^k\|_2}{\|\boldsymbol{W}_t^k\|_2}\frac{\|h^{k-1}(\boldsymbol{x}) - h_{t+\Delta t}^{k-1}(\boldsymbol{x})\|_2}{\|h_t^{k-1}(\boldsymbol{x})\|_2}\right)$$

$$\leq \mu_t c_t \left(\gamma m^{-\beta}\Delta t + \lambda_t \frac{\|h^{k-1}(\boldsymbol{x}) - h_{t+\Delta t}^{k-1}(\boldsymbol{x})\|_2}{\|h_t^{k-1}(\boldsymbol{x})\|_2}\right) \tag{22}$$

We used the sub-multiplicative property of the matrix 2-norm to derive Equation (19), and 1-Lipschitz property of $\phi$ to derive Equation (20). Moreover, Equation (21) comes from Assumption 1 and the definition of $\lambda_t$.

Note that $\frac{\|h_t^0(\boldsymbol{x}) - h_{t+\Delta t}^0(\boldsymbol{x})\|_2}{\|h_t^0(\boldsymbol{x})\|_2} = \frac{\|\boldsymbol{x} - \boldsymbol{x}\|_2}{\|\boldsymbol{x}\|_2} = 0$.

Let $A_k = \max_{\boldsymbol{x} \in X_t} \frac{\|h_t^k(\boldsymbol{x}) - h_{t+\Delta t}^k(\boldsymbol{x})\|_2}{\|h_t^k(\boldsymbol{x})\|_2}$. Then, from the above inequality, we get

$$A_k \leq \mu_t c_t \lambda_t \left(A_{k-1} + \frac{\gamma}{\lambda_t}m^{-\beta}\Delta t\right)$$

$$\leq \mu_t c_t \lambda_t \left(\mu_t c_t \lambda_t \left(A_{k-2} + \frac{\gamma}{\lambda_t}m^{-\beta}\Delta t\right) + \frac{\gamma}{\lambda_t}m^{-\beta}\Delta t\right)$$

$$\vdots$$

$$\leq \left(\frac{\gamma}{\lambda_t}m^{-\beta}\Delta t\right)\sum_{i=1}^{k}(\lambda_t \mu_t c_t)^i$$

where the last inequality holds since $A_0 = 0$

$\square$

Substituting $\Delta t = \Delta t_{\text{sat}}$, $\omega_t^k(\Delta t_{\text{sat}}) = \sqrt{2} + 1$, we get our final theorem.

**Theorem 5.** *Suppose Assumption 3 holds. Let $\lambda_{t,t'}^k = \frac{\|\boldsymbol{W}_{t'}^k\|_2}{\|\boldsymbol{W}_t^k\|_2}$ denote the ratio of the spectral norms of the $k$-th layer weight matrices for task indices $t$ and $t'$, and $\lambda_t = \max_{i \in [L], t' \in [N]} \lambda_{t,t'}^i$. Then, the convergence rate $r_t^k$ of the $k$-th layer representation discrepancy of task $t$ is bounded as*

$$r_t^k \leq (\sqrt{2} - 1)\gamma m^{-\beta}\sum_{i=1}^{k}\frac{(\lambda_t \mu_t c_t)^i}{\lambda_t}, \tag{23}$$

*where $\gamma, \beta$ are positive constants defined in Assumption 3, and $\mu_t, c_t$ are constants defined in Definitions 5 and 6, which depend on the dataset $\mathcal{D}_t$ and the model $h_t$.*

*Proof.* By Thm. 4, we have

$$\omega_t^k(\Delta t) \leq \Big( \frac{\gamma}{\lambda_t} m^{-\beta} \Delta t \Big) \sum_{i=1}^{k} (\lambda_t \mu_t c_t)^i.$$

Therefore

$$
\begin{aligned}
r_t^k &= \frac{1}{\Delta t_{\text{sat}}} \\
&\leq \frac{1}{\omega_t^k(\Delta t_{sat})} \Big( \frac{\gamma}{\lambda_t} m^{-\beta} \Big) \sum_{i=1}^{k} (\lambda_t \mu_t c_t)^i \\
&= \frac{1}{\sqrt{2}+1} \Big( \frac{\gamma}{\lambda_t} m^{-\beta} \Big) \sum_{i=1}^{k} (\lambda_t \mu_t c_t)^i \\
&= (\sqrt{2}-1) \Big( \frac{\gamma}{\lambda_t} m^{-\beta} \Big) \sum_{i=1}^{k} (\lambda_t \mu_t c_t)^i
\end{aligned}
$$

$\square$

# B. Experiments details

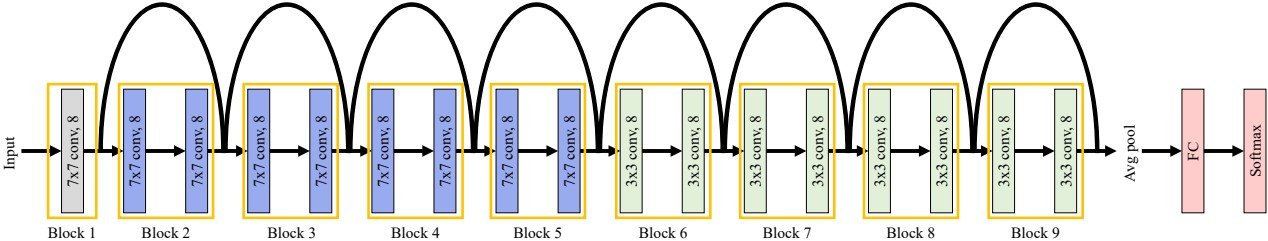

*Figure 7.* ResNet model architecture illustration.

We illustrate the ResNet model used in this paper. Figure 7 depicts the network, which consists of nine blocks. The first block contains a single convolutional layer, while the remaining blocks each have two convolutional layers with residual connections. All convolutional layers use the same channel size, $m = 8$, as the default setting. Specifically, the second through fifth blocks employ a $7 \times 7$ kernel, while blocks six through nine use a $3 \times 3$ kernel. The network concludes with an average pooling layer, followed by a fully connected layer and a softmax output. Notably, we use a (*kernel size*, *stride*, *padding*) triplet of $(7, 1, 3)$ for the larger kernels and $(3, 1, 1)$ for the smaller ones, ensuring the feature map dimensions remain unchanged. Table 1 lists the hyperparameters used for continual learning setups on ImageNet1K and Split-CIFAR100 training.

*Table 1.* Continual learning hyper-parameter for ImageNet32 training

| PARAMETER | VALUE |
|---|---|
| LEARNING RATE | 0.001 |
| BATCH SIZE | 512 |
| EPOCHS | 500 |
| WARM UP STEPS | 200 |
| WORKERS | 4 |
| OPTIMIZER | ADAMW |
| WEIGHT DECAY | 5E-4 |

## B.1. Relationship Between Representation Discrepancy and Representation Forgetting

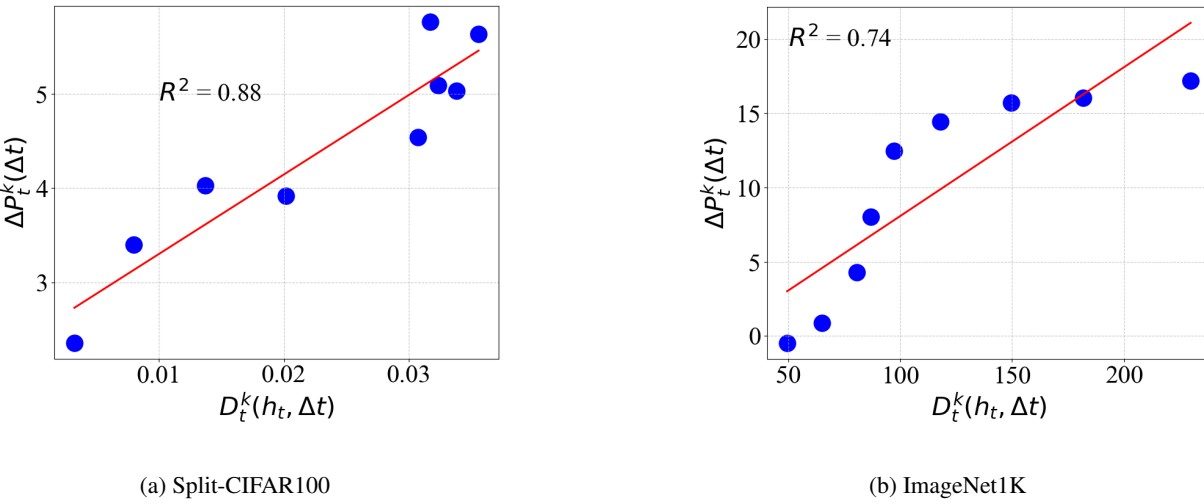

(a) Split-CIFAR100

(b) ImageNet1K

*Figure 8.* Correlation between representation discrepancy $D_t^k(h_t, \Delta t)$ and representation forgetting $\Delta P_t^k(\Delta t)$ on (a) Split-CIFAR100 and (b) ImageNet1K. The strong linear trends (with $R^2$ values of 0.88 and 0.74 respectively) demonstrate that the proposed representation discrepancy metric $D_t^k(h_t, \Delta t)$ serves as an effective surrogate for measuring representation forgetting $\Delta P_t^k(\Delta t)$ in continual learning.

Figure 8 presents empirical validation of the proposed representation discrepancy $D_t^k(h_t, \Delta t)$ as an effective surrogate for representation forgetting. On both Split-CIFAR100 and ImageNet1K, we observe a strong linear relationship between $D_t^k(h_t, \Delta t)$ and $\Delta P_t^k(\Delta t)$, with coefficient of determination $R^2 = 0.88$ and $R^2 = 0.74$, respectively. This high degree of correlation empirically supports that our proposed $D_t^k(h_t, \Delta t)$ can be considered as a practical surrogate for the representation forgetting $\Delta P_t^k(\Delta t)$.

## B.2. Relationship Between Representation Discrepancy and Its Upper Bound

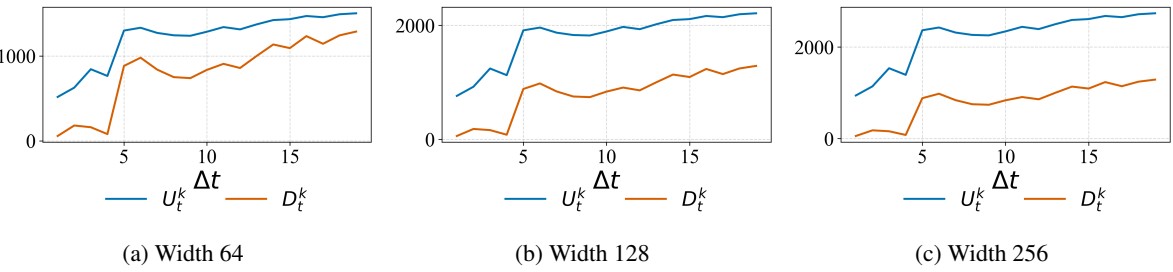

(a) Width 64

(b) Width 128

(c) Width 256

*Figure 9.* Empirical comparison between the theoretical upper bound $U_t^k$ and the measured representation discrepancy $D_t^k$ across ReLU networks of width 64, 128, and 256 on Split-CIFAR100.

Figure 9 presents a comparison between the theoretical upper bound $U_t^k$ and the empirically measured representation discrepancy $D_t^k$, evaluated on ReLU networks trained on the Split-CIFAR100 dataset. We report the results for networks with varying widths $m \in \{64, 128, 256\}$ at layer $k = 5$. Although $U_t^k$ is analytically derived, we observe that $U_t^k$ consistently captures the trend of $D_t^k$ across various setup. This suggests that $U_t^k$ provides a meaningful insight of the representation forgetting dynamics in practice.

## B.3. Comparison of Model Trained on the First Task and Randomly Initialized Model

We assess how far the model is from a theoretical upper bound of forgetting. In Figure 10, we visualize the layerwise representation discrepancy $D_t^k$ between the model trained only on the first task ($h_1$) and two baselines: a randomly initialized model ($h_0$, *i.e.,*, $\Delta t = -1$) and the final model after training on all tasks ($h_N$, *i.e.,*, $\Delta t = N-1$). We observe that across

all layers, the discrepancy $D_t^k(h_1, h_N)$ is consistently and significantly smaller than $D_t^k(h_1, h_0)$. This result implies that the final model retains substantial information about the first task and that the observed saturation in forgetting does not reflect a total erasure of prior knowledge. Therefore, the model has not yet reached a state of *maximum forgetting*, suggesting there is still residual task-specific information preserved even after continual learning.

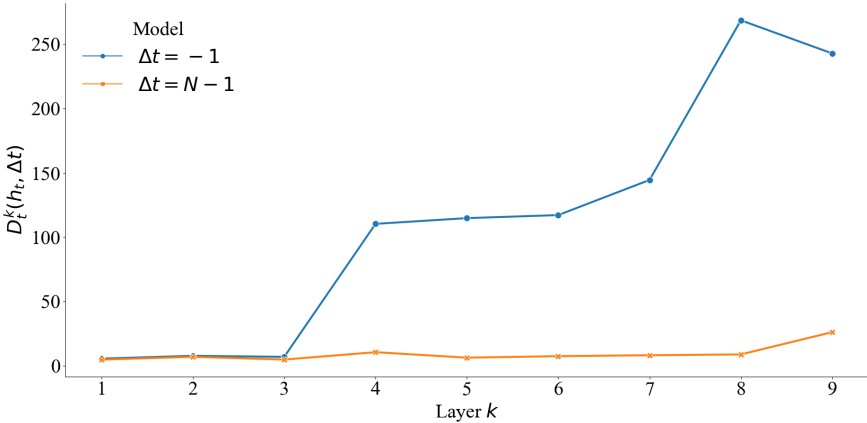

*Figure 10.* Discrepancy between the task-1 model and the final model ($\Delta t = N-1$) remains far below that of a randomly initialized model ($\Delta t = -1$), indicating that the model retains nontrivial task-1 information and has not undergone maximal forgetting.

### B.4. Additional Supplementary Experiments For Assumption 1

We evaluate the validity of Assumption 1 for ResNet by analyzing the flattened weights of the second convolutional layers in each block. Specifically, the weights, originally shaped as $C_o \times C_i \times W \times H$, are reshaped into $C_o \times (C_i \cdot W \cdot H)$, where $C_o$ and $C_i$ represent the output and input channel dimensions, and $W$ and $H$ are the kernel's width and height. To test the assumption, we train a linear model to transform the flattened weights $W_{t'}^k$ into $W_t^k$ for each block $k$. The model is optimized using the AdamW optimizer with a learning rate of $10^{-2}$ and no weight decay. As shown in Figure 2, for every block $k$, we successfully identify a linear transformation $T$ such that $\|W_1^k - TW_{t'}^k\|_2^2$ is minimized. Our results focus on the case where $t = 1$, $t' = N$, and $k \in \{1, \ldots, 9\}$.

Figure 11 illustrates the generality of Assumption 1 in the context of transformer-based architectures. We conduct experiments on a Vision Transformer (ViT) with 9 transformer layers and a single attention head, trained on the Split-CIFAR100 dataset with $N = 50$ tasks. For each layer $k \in \{1, \ldots, 9\}$, we extract the weight matrices from the *MLP block* of the transformer at two checkpoints: after task $t = 1$ and after task $t' = 50$. A linear transformation $T \in \mathbb{R}^{d \times d}$ is then trained to align these weights by minimizing the alignment loss $\|W_t^k - TW_{t'}^k\|_2^2$, averaged over 10 random seeds. We observe that the loss rapidly decreases and approaches zero within 50 epochs across all layers, further supporting the validity of the assumption in ViT models.

### B.5. Additional Experiments on ImageNet1K

In this section, we report the results when we use the ImageNet32 datasets (Chrabaszcz et al., 2017), downsampled from the original ImageNet, to design a series of tasks denoted as $\{\mathcal{D}_i\}_{i=1}^N$. Each task $\mathcal{D}_i$ includes 5 classes selected based on their original class indices. Our setup consists of the first 50 tasks in this ordered split, resulting in $N = 50$ tasks. We train a ResNet (He et al., 2016) with modifications to architectural parameters, including stride and kernel size, to ensure consistent latent feature dimensions. The network comprises 9 blocks ($L = 9$) as shown in Fig. 7 in Appendix.

We observe a strong linear relationship between the representation forgetting $\Delta P_t^k(\Delta t)$ and $\|\mathcal{R}_t^k(h_t)\|$. Here, we set $t = 1$ and $t' = N$ focusing on investigating layers $k \in \{1, .., 9\}$. $\|\mathcal{R}_t^k(h_t)\|$ is computed as the Frobenius norm of the activated output of the residual blocks, while $\Delta P_t^k(\Delta t)$ represents the linear probing accuracy difference between $h_t^k$ and $h_{t'}^k$. In Fig. 12a, $\Delta P_t^k(\Delta t)$ increases proportionally with $\|\mathcal{R}_t^k(h_t)\|$. The linear regression fit (red line) confirms this correspondence, achieving a coefficient of determination $R^2$ of 0.84. Additionally, Figure 12b reveals that $\|\mathcal{R}_t^k(h_t)\|$ scales proportionally with the layer index $k$.

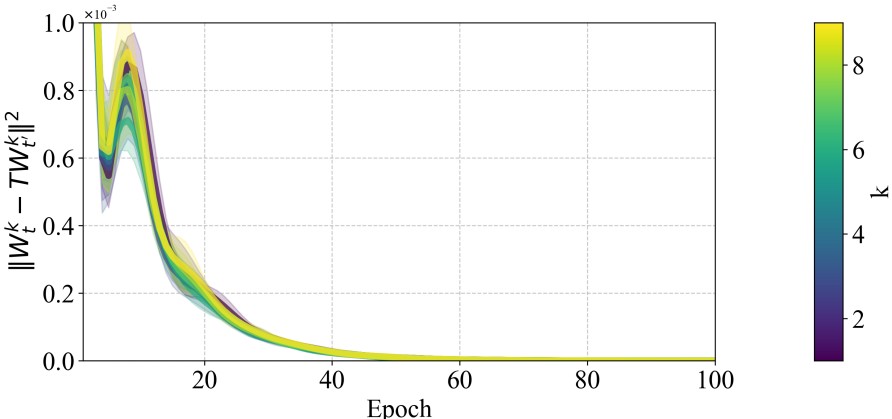

*Figure 11.* Difference between $\boldsymbol{W}_t^k$ and $\boldsymbol{TW}_{t'}^k$ across epochs for each transformer layer $k$ in a Vision Transformer (ViT) trained on Split-CIFAR100. The results show rapid convergence of the linear transformation $\boldsymbol{T}$, validating Assumption 1 in the transformer setting.

### B.6. Analyzing the Effect of $k$ And $m$ on Representation Forgetting

We evaluate the impact of $k$ and $m$ on the convergence rate in Sec. 6.3. Using the hyperparameters listed in Tab. 1 in Appendix, we train a ResNet model on the Split-CIFAR100 dataset. For $k \in \{1, ..., 9\}$ and $m \in \{8 \times 32 \times 32, 16 \times 32 \times 32, 32 \times 32 \times 32\}$, Fig. 13 in Appendix shows the evolution of $\Delta P_t^k(\Delta t)$ as $\Delta t$ grows for $t = 1$. The results confirm that as $k$ increases, $\Delta t_{\text{sat}}$ decreases, indicating a faster convergence rate. Similarly, for varying $m$ and $k = L$, Figure 14 in Appendix shows the evolution of $\Delta P_t^k(\Delta t)$ as $\Delta t$ grows for $t = 1$. The results indicate that as $m$ increases, $\Delta t_{\text{sat}}$ increases, implying a slower convergence rate.

## C. Experimental results on the Representation Discrepancy

### C.1. Evolution of $U_t^k$

We observe that the empirically measured $U_t^k$ stays within the bounded region. We empirically measure $U_t^k(\Delta t)$ on the Split-CIFAR100 and ImageNet1K datasets for $k = L$ and $t = 1$. As shown in Figure 15, the solid blue line represents $U_t^k(\Delta t)$, while the red dashed line denotes the 4-th degree polynomial that best fits the empirical curve, used to visualize the overall trend of representation forgetting. We segment the graph into two regions based on the point where the first local maximum of the fitted polynomial occurs. This point is denoted as $\Delta t_{\text{sat}} = 10$ for Split-CIFAR100 and $\Delta t_{\text{sat}} = 15$ for ImageNet1K, as indicated by the vertical black dashed lines. Note that the values of $U_t^k(\Delta t)$ are shifted such that $U_t^k(0) = 0$, *i.e.*, all values are normalized by subtracting the value at $\Delta t = 0$ to enable consistent comparison across datasets. The shape of the plot for $U_t^k$ is similar to the shape of the upper bound $\Delta P_t^k$ of the representation discrepancy, as shown in Figure 4.

### C.2. Empirical Validation of $U_t^k$ And $\mathcal{R}_t^k$

We examine the behavior of the upper bound $U_t^k$, which captures a tractable proxy for representation discrepancy. In Figure 16, we plot $U_t^k$ against $\|\mathcal{R}_t^k(h_t)\|$ for each layer $k$, using the Split-CIFAR100 and ImageNet1K datasets, respectively. For both datasets, we observe a strong linear correlation, as reflected in high $R^2$ values (0.97 and 0.99), which empirically supports the proportionality $U_t^k \propto \|\mathcal{R}_t^k(h_t)\|$ established in Corollary 1.

### C.3. Convergence Analysis With $U_t^k$

Recall that we empirically measured the convergence rate using the representation forgetting metric $\Delta P_t^k$, as shown in Figure 6. Specifically, we plotted the temporal evolution of $\Delta P_t^k$ for different layer indices $k$, fit a 4-th degree polynomial to each curve, and estimated $\Delta t_{\text{sat}}$ as the time corresponding to the local maximum of the fitted polynomial.

Using the same methodology, we observed a similar trend when measuring convergence with $U_t^k$. As shown in Figure 17, $\Delta t_{\text{sat}}$ consistently decreases as the layer index $k$ increases, suggesting that higher layers converge more quickly. The alignment of results based on both $\Delta P_t^k$ and $U_t^k$ reinforces the conclusion that deeper layers tend to stabilize—and thus

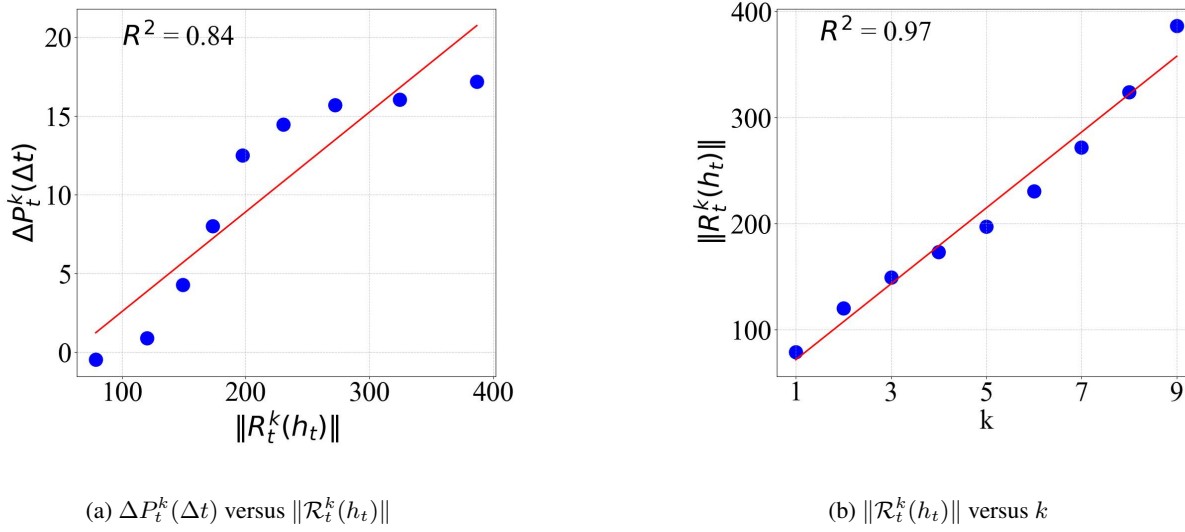

(a) $\Delta P_t^k(\Delta t)$ versus $\|\mathcal{R}_t^k(h_t)\|$            (b) $\|\mathcal{R}_t^k(h_t)\|$ versus $k$

*Figure 12.* $\|\mathcal{R}_t^k(h_t)\|$ explains the representation forgetting $\Delta P_t^k(\Delta t)$. Each point represents each layer from the ResNet. In Fig 12a, the red line shows the linear regression fit between $\|\mathcal{R}_t^k(h_t)\|$ and $\Delta P_t^k(\Delta t)$, demonstrating a strong correlation with $R^2 = 0.84$. In Fig. 12b, $\|\mathcal{R}_t^k(h_t)\|$ is shown to increase proportionally with the layer index $k$. These results indicate that representation forgetting, $\Delta P_t^k(\Delta t)$, is closely linked to both $\|\mathcal{R}_t^k(h_t)\|$ and the layer index $k$.

forget—more rapidly than shallower ones.

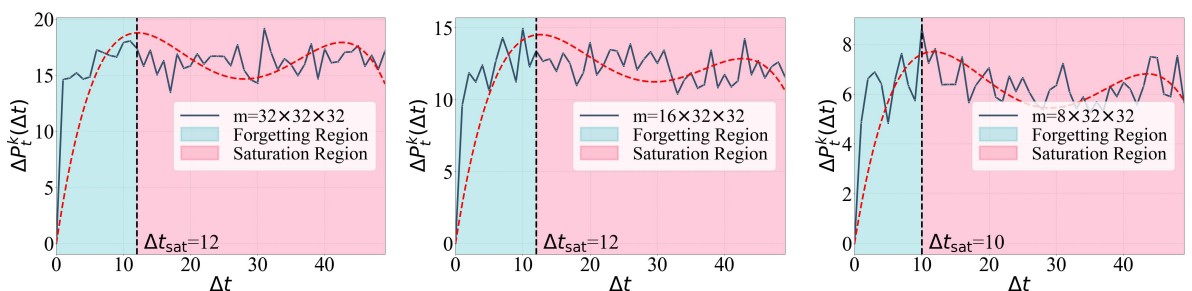

*Figure 13.* The impact of $k$ on the convergence of representation discrepancy.

*Figure 14.* The impact of $m$ on the convergence of representation discrepancy.

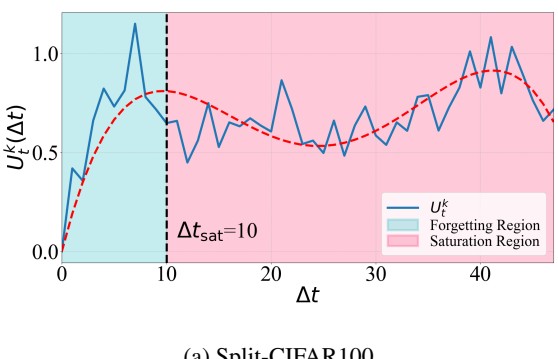

(a) Split-CIFAR100

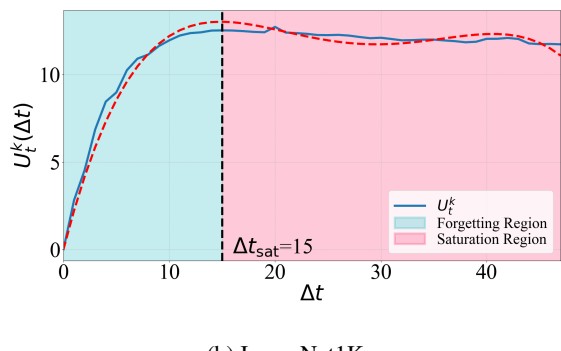

(b) ImageNet1K

*Figure 15.* Evolution of the empirically measured upper bound $U_t^k(\Delta t)$ on the representation discrepancy for $t = 1$ and $k = L$, evaluated on (a) Split-CIFAR100 and (b) ImageNet1K. The solid blue line shows $U_t^k(\Delta t)$, and the red dashed line indicates the 4-th degree polynomial fit used to reveal the overall trend of forgetting. Each graph is divided into two phases, separated at $\Delta t_{\mathrm{sat}} = 10$ for Split-CIFAR100 and $\Delta t_{\mathrm{sat}} = 15$ for ImageNet1K, corresponding to the first local maximum of the polynomial fit. All curves are normalized such that $U_t^k(0) = 0$ for consistent comparison.

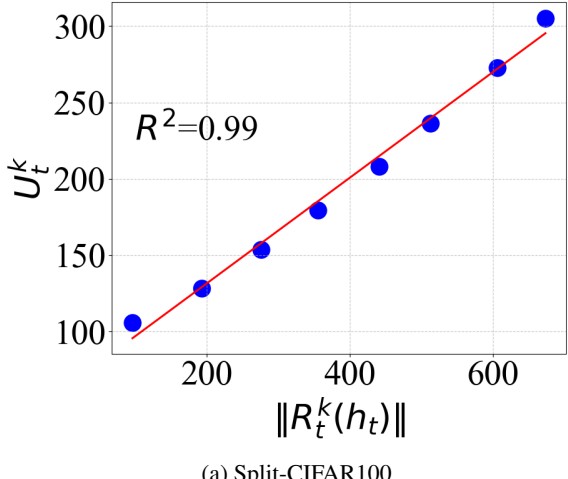

(a) Split-CIFAR100

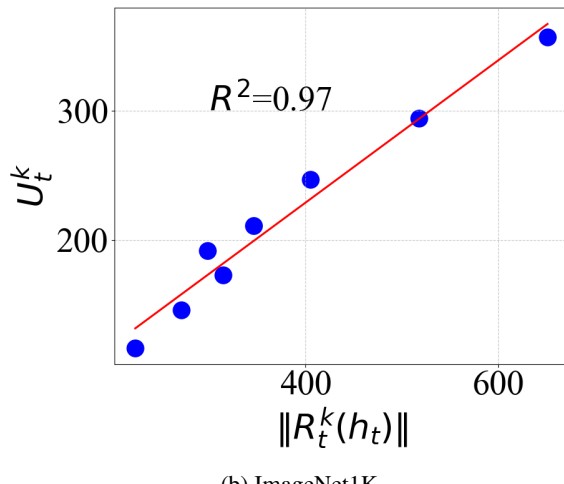

(b) ImageNet1K

*Figure 16.* Relationship between the $U_t^k$ and $\|\mathcal{R}_t^k(h_t)\|$, across layers $k = 1, \ldots, 9$, for Split-CIFAR100 (left) and ImageNet1K (right). Each point corresponds to a specific layer. The fitted red lines and high $R^2$ values (0.97 and 0.99) confirm a strong linear relationship, supporting the proportionality $U_t^k \propto \|\mathcal{R}_t^k(h_t)\|$ stated in Corollary 1.

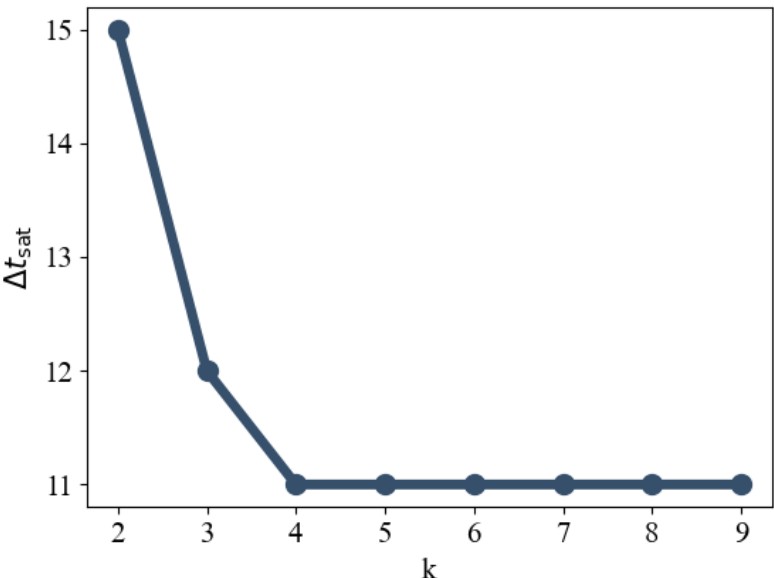

*Figure 17.* Saturation time $\Delta t_{\text{sat}}$ measured using $U_t^k$ for different layer indices $k$. Following the same methodology as in Figure 6, we fit a 4-th degree polynomial to the temporal trajectory of $U_t^k$ and estimate $\Delta t_{\text{sat}}$ as the time corresponding to the local maximum of the fitted curve. The result shows that $\Delta t_{\text{sat}}$ decreases with increasing $k$, indicating faster convergence in deeper layers.

