# OpenReview forum: "Measuring Representational Shifts in Continual Learning: A Linear Transformation Perspective"
_ICML.cc/2025/Conference — ICML 2025 poster_

### Official Review · Reviewer_cno5 · 2025-02-27

**Overall Recommendation:** 4

**Summary:**

This paper focuses on the theoretical analysis of representation forgetting in Continual Learning scenarios. While representation forgetting has been introduced previously, the focus was largely experimental and the main contribution of the proposed work is to focus on the theoretical aspect of representation forgetting. Therefore, the authors justify the usage of a new metric, Representation Decrepancy, as a surrogate for traditional representation forgetting measures. Consequently, this paper analyzes the theoretical behavior of representation forgetting, showing its dependence on layer index, and network depth, and identifying various regimes of representation forgetting during training. Such findings are supported by experiments.

## update after rebuttal

I thank the authors for their detailed responses and clarifications.

The figure given in response R4-3 is interesting and I would suggest including it in the final version of the manuscript (or in appendix).

Regarding R4-2, I should have been more precise in my comment, but I believe the main experimental result of the paper lies in Figure 4, which is what I would like to see regarding the evolution of $D^k_t$. Although as stated, I understand that $D^k_t$ and $\Delta P^k_t$ are correlated so similar results should be observed. Nonetheless, the authors should display the results with $D^k_t$ since the theoretical analysis is made with this quantity.

For these reasons, I will maintain my current score of 4.

**Claims And Evidence:**

The theoretical aspect of the paper is clear and adequately justified. I did not find any major flow in the presented theorems or assumptions.

My main concern resides in the experimental section. While the author introduces a new metric, representation discrepancy, they somehow never compute it in the experiments to the benefit of representation forgetting as defined in previous work. While the authors indeed discuss how both metrics are related, I struggle to understand the justification of such a choice and I am unsure that this adequately justifies the theoretical claims.

**Essential References Not Discussed:**

The increase of forgetting in latest layers is intuitive and expected. This has similarly been discussed in previous studies such as [1]. While the theoretical findings are interesting, further discussion with previous studies would be included.

[1] Ramasesh, Vinay Venkatesh, Ethan Dyer, and Maithra Raghu. "Anatomy of Catastrophic Forgetting: Hidden Representations and Task Semantics." _International Conference on Learning Representations_.

**Experimental Designs Or Analyses:**

The experimental design is clearly explained and overall makes perfect sense for demonstrating the theoretical claim. As already stated, my main concern remain the metric used in epxeriments.

**Methods And Evaluation Criteria:**

Yes, the dataset perfectly makes sense, even though additional settings would be considered. However, the paper being mostly theoretical, I believe the current choice to be sufficient.

**Other Comments Or Suggestions:**

- The increase of forgetting in the latest layers is intuitive and expected. This has similarly been discussed in previous studies such as [1]. While the theoretical findings are interesting, further discussion with previous studies would be included.

- From a theoretical perspective, it seems to me that such analysis can be adequately applied beyond Class-Incremental Learning scenarios (which is the only one considered here). I would appreciate discussion in this regard. While I believe the current experiments to be sufficient, experiments in Domain Incremental Learning cases would further strengthen the theoretical findings.

**Other Strengths And Weaknesses:**

*Strenghts*

- The writing quality is very good, I enjoyed reading the paper and found the presentation particularly clear
- The findings are interesting and novel
- Illustrations are relevant
- The presence of experiments to support the theoretical claims is particularly appreciated

*Weaknesses*
- I would like the code to be shared
- See questions and suggestions

**Questions For Authors:**

- What about when the representations are normalized? Such strategy is very common in representation learning, how would the current analysis be impacted?
- Why not use centroid distances for definition 3, instead of the most distant features?
- How far is the theoretical upper bound $U_{t, \infty}^k$ from an experimental upper bound?  For example, what is the discrepancy between the model trained on the first task and a randomly initialized model? Such information could be interesting to understand how much the model *can* forget, as for now it is not clear if this saturation is the consequence of "not being able the forget more" or if it occurs before reaching such a stage, therefore maintaining some of the previous knowledge (and if so, how far are we from "maximum forgetting"?).
- My biggest concern is the following. In Section 6.1, why compute the representation forgetting $\Delta P_t^k(\Delta t)$  as defined by [2], instead of directly computing your newly introduced metric $D_t^k(h_t,\Delta t)$? I understand that the two are correlated as stated in 4.2 but all the theoretical analysis is made on $D_t^k(h_t,\Delta t)$, which should be computable. I am unsure how much should the analysis of $\Delta P_t^k(\Delta t)$ gives insight on the behavior of the maximum value of $D_t^k(h_t,\Delta t)$. Could you provide more explanation on this choice and its consequences?

**Relation To Broader Scientific Literature:**

This is related to Continual Learning in general as it helps understand how neural network representation can change throughout training. Notably, while some findings have been observed experimentally in previous studies (forgetting strong in the latest layers), it gives additional insight into being able to connect it with the theory proposed in the paper. Maybe such theoretical analysis can help design new methods for reducing forgetting in intermediate layers or give additional metrics for quantifying training behaviors.

**Theoretical Claims:**

I checked the correctness of the main paper, however only briefly looked at the demonstrations in the appendix. So far, no major issues has been identified.

---

> ### Author Rebuttal · Authors · 2025-03-31
>
> We thank the reviewer cno5 for the detailed review and constructive suggestions. We appreciate your acknowledgements that the theoretical aspects of this paper are **clear and adequately justified**, and that the findings are **interesting and novel**. Below we show our reply on your comments and questions.
>
> ---
>
> **[R4-1]**`In Section 6.1, why compute the representation forgetting \Delta P^k_t instead of directly computing your newly introduced metric D^k_t? I understand that the two are correlated as stated in 4.2 but all the theoretical analysis is made on D^k_t, which should be computable`
>
> As noted in [R3-3] and [R3-6], we evaluated $D_t^k$ directly and observed results consistent with Figures 5 and 6. These will be included in the revised manuscript.
>
> ---
>
> **[R4-2]**`I am unsure how much should the analysis of \Delta P^k_t gives insight on the behavior of the maximum value of D^k_t. Could you provide more explanation on this choice and its consequences?`
>
> Thank you for your insightful comments. Our original intent in conducting experiments with $\Delta P^k_t$ was to demonstrate that the theoretical results derived from $D^k_t$ also holds in $\Delta P^k_t$. However, we agree that experiments based solely on $\Delta P^k_t$ are insufficient to support the theoretical claims concerning $D^k_t$. Accordingly, we have conducted additional experiments using $D^k_t$, as detailed in [R3-3] and [R3-6].
>
> ---
>
> **[R4-3]**`How far is the theoretical upper bound U^k_t,\infty from an experimental upper bound? For example, what is the discrepancy between the model trained on the first task and a randomly initialized model? Such information could be interesting to understand how much the model can forget, as for now it is not clear if this saturation is the consequence of "not being able the forget more" or if it occurs before reaching such a stage, therefore maintaining some of the previous knowledge (and if so, how far are we from "maximum forgetting"?).`
>
> We thank the reviewer for the insightful question. To clarify, at each layer $k$, the below plot reports the representation discrepancy $D_t^k$ between the task-1 model $h_1$ and two models:
> * a randomly initialized model $h_0$, i.e., $\Delta t = -1$
> * the final model after training on all tasks $h_N$, i.e., $\Delta t = N-1$
>
> Our results show that $D^k_1(h_1, N-1)$ is significantly smaller than $D^k_1(h_1, -1)$, indicating that the saturation does not correspond to complete forgetting. Thus, the model retains a nontrivial amount of task-1 information even after learning all subsequent tasks.
>
> https://hackmd.io/_uploads/ByCfAQOTke.png
>
> ---
>
> **[R4-4]**`The increase of forgetting in latest layers is intuitive and expected. This has similarly been discussed in previous studies such as [1]. While the theoretical findings are interesting, further discussion with previous studies would be included.`
>
> Thank you for the suggestion.
>
> We will include a discussion comparing our results with [1], which empirically observed greater drift in deeper layers. Our work complements this by providing the first theoretical explanation of this phenomenon (see Corollary 1 and Fig. 5).
>
> ---
>
> **[R4-5]**`I would like the code to be shared`
>
> You can find code here: [link](https://anonymous.4open.science/r/representational-shifts-in-cl-F1F5/README.md)
>
> ---
>
> **[R4-6]**`From a theoretical perspective, it seems to me that such analysis can be adequately applied beyond Class-Incremental Learning scenarios (which
> is the only one considered here). I would appreciate discussion in this regard. While I believe the current experiments to be sufficient, experiments in Domain Incremental Learning cases would further strengthen the theoretical findings.`
>
> We agree that our framework can extend to domain-incremental learning (DIL), provided an appropriate modeling of weight perturbation as done in Assumption 2 for class-incremental learning. While a formal theory for DIL is left for future work, we provide preliminary empirical evidence in [R2-2] using rotated Split-CIFAR100, where Corollary 1 continues to hold.
>
> ---
>
> **[R4-7]**`What about when the representations are normalized? Such strategy is very common in representation learning, how would the current analysis be impacted?`
>
> While we do not yet have a formal analysis under normalization, we offer the following intuition: normalization compresses the representation space by reducing feature norms, which in turn may reduce the magnitude of $D_t^k$. If the shift in linear structure is preserved under normalization, our discrepancy measure would still reflect representational drift, but with smaller scale. We will mention this in the revised manuscript.
>
> ---
>
> **[R4-8]**`Why not use centroid distances for definition 3, instead of the most distant features?`
>
> We appreciate the suggestion. We follow the max-discrepancy formulation as in Guha et al. (2024), who adopted a similar worst-case approach in their theoretical analysis of catastrophic forgetting.

---

> > ### Comment · Reviewer_cno5 · 2025-04-04
> >
> > I thank the authors for their detailed responses and clarifications.
> >
> > The figure given in response R4-3 is interesting and I would suggest including it in the final version of the manuscript (or in appendix).
> >
> > Regarding R4-2, I should have been more precise in my comment, but I believe the main experimental result of the paper lies in Figure 4, which is what I would like to see regarding the evolution of $D^k_t$. Although as stated, I understand that $D^k_t$ and $\Delta P^k_t$ are correlated so similar results should be observed. Nonetheless, the authors should display the results with $D^k_t$ since the theoretical analysis is made with this quantity.
> >
> > For these reasons, I will maintain my current score of 4.

---

> > > ### Author Response · Authors · 2025-04-07
> > >
> > > We thank the reviewer for maintaining a positive assesment of our work. Following the suggestion, we will include the figure presented in R4-3 in the final version of our manuscript. We also agree that it is important to report the empirical evolution of D^k_t. Accordingly, we will include the corresponding results in the final version as well.

---

### Official Review · Reviewer_4iQx · 2025-03-09

**Overall Recommendation:** 3

**Summary:**

In this works, the authors introduce a novel measure of forgetting in the hidden layers of deep neural networks in continual learning setting. They derive an upper-bound for the proposed representation discrepancy measure and the convergence rate of this measure under a set of assumptions. Additionally, the authors evaluate the development of representation forgetting during learning empirically using Split-CIFAR100 and ImageNet 1K.

**Claims And Evidence:**

1. The manuscript doesn’t clearly justify how the proposed measure of representation discrepancy $D^k\_t$ relates to forgetting, despite claiming it as an effective surrogate for representation forgetting. While the authors demonstrate that when $D^k\_t$ is small, $P^k\_t$ is also small, it remains unclear how these values scale or whether one provides an upper bound for the other. This is particularly concerning, because the empirical results evaluate the properties of only $P^k\_t$, not $D^k\_t$. Furthermore, the convergence rate in Fig. 6 is defined using $P^k\_t$, which is different from the original definition in Eq. 7. Thus, I find that the analytical results do not support their claims on forgetting convincingly, and there remains discrepancy between analytical and empirical findings.

2. In the proof of proposition 1, the authors state that $d$ increases linearly with $\Delta t$, citing Theorem 4.1 of Guha & Lakshman (2024). However, in the original theorem, this linear $\Delta t$ dependence is shown for the upper bound of $d$, not $d$ itself, making this proof and the following arguments on the two-stage dynamics not well supported.

3. The motivation behind the definition of the representation discrepancy is unclear. Because the definition relies on the maximum discrepancy over all samples rather than the distance between two activity distributions, it is susceptible to outlier, raising concerns about its robustness.

4. In section 4.1, the introduction of a new measure for representation forgetting is motivated by the difficulty of deriving an optimal linear classifier for hidden layers. I’m confused by this argument particularly because the proposed measure of representation discrepancy requires derivation of the optimal linear transformation between two hidden layer representations, which is typically computationally heavier than the derivation of the optimal linear classifier (if $d_y < w_k$).

**Essential References Not Discussed:**

Not to my knowledge.

**Experimental Designs Or Analyses:**

Yes.

**Methods And Evaluation Criteria:**

See the comment 1 above.

**Other Comments Or Suggestions:**

Naively thinking, Assumption 1 shouldn't hold when $w_k < w\_{k-1}$, but Figure 2 suggests it does. Why is that the case?

**Other Strengths And Weaknesses:**

The question addressed in this manuscript, forgetting in hidden layers of ANNs, is potentially interesting and important.

**Questions For Authors:**

Please see the comments above.

**Relation To Broader Scientific Literature:**

While the manuscript is built heavily upon Guha & Lakshman (2024), the results presented here are clearly distinct from that previous work.

**Theoretical Claims:**

See the comment 2 above.

---

> ### Author Rebuttal · Authors · 2025-03-31
>
> We thank the reviewer 4iQx for the detailed review and constructive suggestions. We appreciate your acknowledgements that the problem addressed in this manuscript is potentially **important and interesting**, and that our results are **clearly distinct from the previous works**. Below we show our reply on your comments and questions.
>
> ---
> **[R3-1]**`The introduction of a new measure for representation forgetting is motivated by the difficulty of deriving an optimal linear classifier for hidden layers. I’m confused by this argument.`
>
> We believe that there may have been a misunderstanding regarding our motivation for introducing the measure $D^k_t$.
>
> The term *difficulty* in our statement refers to theoretical intractability rather than computational cost. Our intention was **not** to use $D^k_t$ as a *more efficient alternative* for computing representation forgetting in practical settings. Rather, we introduced $D^k_t$ to *facilitate the theoretical analysis* of $\Delta P^k_t$.
>
> We will clarify this point in the revised manuscript.
>
> ---
> **[R3-2]**`It remains unclear how D^k_t and P^k_t scale or whether one provides an upper bound for the other. `
>
> Below we added plots showing $D_t^k$ versus $\Delta P_t^k$ across different layers $k$. In both datasets, $\Delta P_t^k$ **scales approximately linearly** with $D_t^k$, supporting its role as a surrogate. We will include these results in the revised manuscript.
>
> | **ImageNet1K**                                           | **SplitCIFAR100**                                         |
> |-|-|
> |https://hackmd.io/_uploads/BJVuwbdp1g.png|https://hackmd.io/_uploads/Sk-YD-upkl.png|
>
> ---
> **[R3-3]** `The empirical results evaluate the properties of only P^k_t not D^k_t. Thus, the analytical results do not support their claims on forgetting convincingly.`
>
> We conducted additional experiments directly evaluating $D_t^k$. As shown below, we observed a **strong linear relationship** between $D_t^k$ and $R_t^k$, similar to our observation in Fig.5 of the submitted manuscript. We also confirm the linearity between $U_t^k$ and $R_t^k$ as predicted by Corollary 1.
>
> These results will be included in the revised manuscript.
>
> | **ImageNet1K**                                           | **SplitCIFAR100**                                         |
> |-|-|
> |https://hackmd.io/_uploads/BJ-jMLEpJg.png|https://hackmd.io/_uploads/S1RsG8Vpye.png|
> |https://hackmd.io/_uploads/SkTYGzd6Je.png|https://hackmd.io/_uploads/B1n9MGOa1l.png|
>
> ---
> **[R3-4]**`The motivation behind the definition of the representation discrepancy is unclear. It is susceptible to outlier, raising concerns about its robustness.`
>
> Thank you for your comment. As noted in [R3-1], our primary motivation for introducing the metric $D^k_t$ is to facilitate the theoretical analysis of $\Delta P^k_t$. We adopted the maximum discrepancy formulation following the approach of Guha et al. (2024), who succesfully employed a similar method in their theoretical study of catastrophic forgetting in continual learning.
>
> ---
> **[R3-5]**`The proof of Prop 1., claims d increases linearly with \Delta t, citing Thm 4.1 of Guha & Lakshman (2024), but the theorem only shows this dependence for the upper bound of d. Hence, the proof and the following arguments on the two-stage dynamics is not well supported.`
>
> Thank you for the sharp comment. The reviewer is correct that the linear dependency of $d$ with respect to $\Delta t$ is not *theoretically* proven in Guha & Lakshman (2024). As such, our proof in Proposition 1 requires an additional assumption (that the lower bound of $d$ increases without a limit as a function of $\Delta t$) for the result to hold.
>
> Under this assumption, the unique peak described in Proposition 1 may not always be guaranteed. However, this assumption is sufficient for the two-phase dynamics to hold, as the upper bound $U^k_t$ will eventually reach a peak and then saturate.
>
> We also note that if $d$ itself monotonically increases with $\Delta t$, then the uniqueness of the peak as stated in Proposition 1 follows. We will clarify these assumptions and their implications in the revised version.
>
> ---
> **[R3-6]**`The convergence rate in Fig. 6 is defined using P^k_t, which is different from the original definition in Eq. 7.`
>
> As suggested, we report the convergence rate using $U_t^k$ instead of $\Delta P_t^k$. Similar to Figure 6, we find that $\Delta t_{\text{sat}}$ decreases with layer index $k$, so the convergence rate $r_t^k = 1 / \Delta t_{\text{sat}}$ increases with $k$, consistent with Theorem 2.
>
> https://hackmd.io/_uploads/r1VnzXOp1l.png
>
> ---
> **[R3-7]**`Naively thinking, Assumption 1 shouldn't hold when w_k < w_{k-1}. Why is that the case?`
>
> Assumption 1 compares across different tasks $t$ and $t'$ for the same layer $k$, not across different layers. Thus, the width $w_k$ remains fixed for each comparison, and the condition holds regardless of whether $w_k < w_{k-1}$. This is why Assumption 1 is satisfied in Figure 2.

---

> > ### Comment · Reviewer_4iQx · 2025-04-04
> >
> > I thank the authors for the rebuttal. With the addition of the numerical evaluation of $D^k_t$ and the clarification of the assumption for Proposition 1, the manuscript is technically sound. I have thus raised my evaluation, although I’m not fully convinced of the utility of the measure based on the maximum discrepancy.
> >
> > Regarding [R3-7], because $W^k \in R^{w_k \times w_{k-1}}$, there exists $T \in R^{w_k \times w_k}$ satisfying $T W^k_{t’} = W^k_t$ almost trivially if $w_k \geq w_{k-1}$. However, the existence of $T$ is not guaranteed if $w_k < w_{k-1}$. I assume it works because learned weights tend to be effectively low-rank.

---

> > > ### Author Response · Authors · 2025-04-07
> > >
> > > We thank the reviewer for the positive evaluation and for acknowledging the technical soundness of the manuscript.
> > >
> > > > Regarding [R3-7], because $W^k \in R^{w_k \times w_{k-1}}$, there exists $T \in R^{w_k \times w_k}$ satisfying $T W^k_{t’} = W^k_t$ almost trivially if $w_k \geq w_{k-1}$. However, the existence of $T$ is not guaranteed if $w_k < w_{k-1}$. I assume it works because learned weights tend to be effectively low-rank.
> > >
> > > We would like to clarify that, as stated in Assumption 1, we consider the case where $w_k = w_{k-1}$ for all $k$. Therefore, all experiments related to Figure 2 were conducted using networks with equal layer widths, i.e, $w_k = w_{k-1}$. We will make this explicit in the revised manuscript to avoid any confusion.

---

### Official Review · Reviewer_9B8C · 2025-03-10

**Overall Recommendation:** 4

**Summary:**

This paper introduces a novel metric—termed representation discrepancy—to quantify the degradation of internal feature representations (i.e., representation forgetting) in continual learning. By framing the forgetting problem in terms of a minimum alignment error between hidden layer representation spaces via an optimal linear transformation, the authors provide both a theoretical analysis and empirical validation.

**Claims And Evidence:**

The primary claim is that the proposed representation discrepancy is a reliable and analytically tractable surrogate for measuring representation forgetting. The authors support this claim through rigorous theoretical derivations (e.g., Theorem 1 and Theorem 2) and by demonstrating a strong empirical correlation between the discrepancy and conventional measures (such as degradation in linear probing accuracy). While the derivations are nontrivial and depend on certain assumptions (e.g., the existence of a linear transformation aligning weight matrices across tasks), the evidence—both theoretical and experimental—is generally convincing. Some aspects, particularly the sensitivity of these assumptions to different network architectures, might benefit from further discussion.

**Essential References Not Discussed:**

To my knowledge, no.

**Experimental Designs Or Analyses:**

The experiments are designed to validate the theoretical predictions: the evolution of representation forgetting is tracked as new tasks are learned, and the relationships between layer depth, network width, and forgetting are examined. The use of visualizations (e.g., plotting the forgetting curves and the relationship between representation space size and forgetting) helps illustrate the key findings. While the experimental setup is solid, additional ablation studies—such as varying network architectures or exploring different continual learning scenarios—could further substantiate the results.

**Methods And Evaluation Criteria:**

The method of aligning representation spaces using a linear transformation is well-motivated and novel in the context of continual learning. The evaluation criteria, which include comparing linear probing performance before and after task transitions, appear appropriate for assessing representation quality. The choice of benchmark datasets and the experimental design are standard for the field, lending credibility to the claims. However, additional details on hyperparameter sensitivity and broader dataset diversity could further strengthen the evaluation.

**Other Comments Or Suggestions:**

The authors can consider providing more intuition on the practical implications of the theoretical findings—especially how this metric could potentially inform better continual learning strategies.

**Other Strengths And Weaknesses:**

Strengths:
1.The paper proposes a novel and theoretically grounded metric for a long-standing problem in continual learning.
2. The theoretical analysis is detailed and offers clear predictions about how factors like layer depth and network width influence forgetting.
3. Experimental validation on standard datasets is convincing and well-presented.

Weaknesses:
1. Some of the core assumptions (e.g., the existence of a near-perfect linear alignment between weight matrices) may not generalize across all architectures or learning scenarios.
2. Certain proofs and derivations could benefit from additional clarity, and more discussion on potential limitations would be helpful.

**Questions For Authors:**

Could you elaborate on the limitations of Assumption 1 regarding the existence of a linear transformation aligning weight matrices? How might this assumption be relaxed or validated for other network architectures (e.g., non-ReLU networks or models with skip connections)?

**Relation To Broader Scientific Literature:**

This work builds on a rich body of literature in continual learning and catastrophic forgetting. It is well situated relative to prior work—especially that of Guha & Lakshman (2024) and other studies addressing representation changes over time. The connection drawn between representation forgetting and linear probing performance is particularly insightful. However, integrating discussion of a few more recent works on unsupervised continual learning and contrastive methods might further contextualize the contributions.

**Theoretical Claims:**

The paper presents several theoretical claims, including explicit upper bounds on the representation discrepancy and an analysis of the convergence rate of forgetting across layers. The proofs, particularly for Theorem 1 and Theorem 2, seem methodologically sound, though they rely on assumptions (such as Assumption 1 regarding the existence of a suitable linear transformation) that, while empirically supported, might not hold in all settings. Clarifying the limitations of these assumptions and their impact on generality would improve the theoretical discussion.

---

> ### Author Rebuttal · Authors · 2025-03-31
>
> We thank the reviewer 9B8C for the detailed review and constructive suggestions. We appreciate your acknowledgements that our theoretical and experimental results are generally **well-presented and convincing**, and that our **proofs are methodologically sound**. Below we show our reply on your comments and questions.
>
> ---
>
> **[R2-1]** `Generalizability and limitations of Assumption 1`
>
> Thank you for raising this important point.
>
> We agree that **Assumption 1** (existence of a near-perfect linear transformation $T$ aligning $W_t^k$ and $W_{t'}^k$) may not universally hold. In the revised manuscript, we clarify its scope and limitations.
>
> As shown in **Figure 2**, Assumption 1 holds empirically for **MLPs** and **ResNets**, the latter of which includes skip connections (asked by the reviewer). To test generality, we conducted experiments on a **Vision Transformer (ViT)** (9 layers, 1 head) trained on **Split-CIFAR100**, optimizing $T$ to align $W_1^9$ and $W_{50}^9$, which showed clear convergence:
>
> https://hackmd.io/_uploads/SkzzjJ4p1x.png
>
> We also compare alignment errors:
>
> | Architecture           | Final Alignment Error         |
> |------------------------|-------------------------------|
> | Multi-Layer Perceptron | $7.58 \times 10^{-7}$         |
> | ResNet                 | $5.18 \times 10^{-7}$         |
> | Vision Transformer     | $6.52 \times 10^{-8}$         |
>
> While an approximate version of the assumption (e.g., $\| W_t^k - T W_{t'}^k \| \leq \varepsilon$) is possible, understanding its effect on theory is left for future work. This limitation will be explicitly discussed.
>
> ---
>
> **[R2-2]** `Ablations across architectures or continual learning scenarios`
>
> Thanks for the suggestion.
>
> We added experiments under a **domain-incremental setup**, training ResNet on **rotated Split-CIFAR100** (same classes, different input rotations). We checked $R_t^k$ vs. $U_t^k$ for $t=1$ and $k=1,\dots,9$, observing consistent linear trends as predicted by Corollary 1:
>
> https://hackmd.io/_uploads/r14Vgb_Tyg.png
>
> In addition, we ran additional experiments for **Vision Transformer (ViT)** architecture to check the validity of Assumption 1, as shown in our response in [R2-1].
>
> ---
>
> **[R2-3]** `Clarifying proofs and discussing limitations`
>
> Thank you. We revised the proofs to add explanations in non-trivial derivations:
>
> | Submitted Version                                      | Revised Version                                        |
> |--------------------------------------------------------|--------------------------------------------------------|
> | https://hackmd.io/_uploads/rkg3RbB61l.png | https://hackmd.io/_uploads/r15TpWr6ke.png  |
> | https://hackmd.io/_uploads/HJmS0WSpyl.png | https://hackmd.io/_uploads/ryHb0bH61g.png |
>
> For discussion on potential limitations of our assumption, please refer to our response in [R2-1].
>
> ---
>
> **[R2-4]** `Referencing recent unsupervised and contrastive learning works`
>
> While our focus is on supervised continual learning, we agree that linking to recent related work adds context. We will include the following:
>
> - Malviya et al., *J. Comput. Sci.*, 2025
> - Wen et al., *arXiv:2405.18756*, 2024
> - Zhang et al., *arXiv:2404.19132*, 2024
>
> ---
>
> **[R2-5]** `Practical implications of the theoretical findings`
>
> Thank you for the insightful suggestion.
>
> While our metric is theoretical in nature, the **insights derived from it** suggest actionable strategies for improving continual learning (see [R1-5]). We will emphasize this connection more clearly in the revision.
>
> ---
>
> **[R2-6]** `Hyperparameter sensitivity and dataset diversity`
>
> Thank you for the constructive suggestion.
>
> We conducted experiments varying **batch size** ($B = 128, 256, 512$) and **learning rate** ($\gamma = 10^{-3}, 10^{-4}, 10^{-5}$) on Split-CIFAR100 using ResNet. The linear relationship between $R_t^k$ and $U_t^k$ remained consistent, demonstrating robustness:
>
> | **Batch size** $B$                                       | **Learning rate** $\gamma$                                  |
> |----------------------------------------------------------|-------------------------------------------------------------|
> | https://hackmd.io/_uploads/H1jp6bOpJx.png | https://hackmd.io/_uploads/HkZ06bOa1e.png |
>
> As for dataset diversity, see [R2-2] for domain-incremental results on **rotated Split-CIFAR100**.

---

> > ### Comment · Reviewer_9B8C · 2025-04-02
> >
> > Thanks for the reply, I've read them but I want to keep my score as 4.

---

> > > ### Author Response · Authors · 2025-04-07
> > >
> > > We thank the reviewer for reading our comments and maintaining the positive evaluation.

---

### Official Review · Reviewer_1dag · 2025-03-14

**Overall Recommendation:** 3

**Summary:**

This paper introduces a novel metric for measuring representation forgetting in continual learning and derives an upper bound for this metric. The theoretical findings provide valuable insights, which are further validated through experiments on real image datasets.

**Claims And Evidence:**

Yes.

**Essential References Not Discussed:**

No.

**Experimental Designs Or Analyses:**

No.

**Methods And Evaluation Criteria:**

Yes.

**Other Comments Or Suggestions:**

NA.

**Other Strengths And Weaknesses:**

Overall, this is a well-written paper with a clearly stated system model and key assumptions. The proposed metric is a reasonable choice, particularly for facilitating theoretical analysis. However, the interpretation of Theorem 1 is insufficient, especially regarding the tightness of its upper bound, which is not discussed.

**Questions For Authors:**

1. How useful are these insights for improving the performance of continual learning?

2. In Definition 3, the self-distance is nonzero. Does this indicate that Definition 3 is not suitable for measuring distances between representation spaces?

3. Without a discussion on the tightness of the upper bound, certain interpretations—such as the peak value in Proposition 1—become questionable. Additionally, other aspects of the paper may also raise concerns about precision and tightness, such as Definitions 3 and 4. Specifically, the proposed distance/discrepancy measures are only approximations based on imprecise beliefs (as indicated by the use of “$\approx$” in Section 4.2).

4. Figures 4 and 8 display multiple peaks, which contradicts the unique peak predicted by Proposition 1. Could this be evidence that the theoretical upper bound is not sufficiently tight?

**Relation To Broader Scientific Literature:**

This paper presents theories in continual learning, which helps the general understanding of lifelong learning systems.

**Theoretical Claims:**

No.

---

> ### Author Rebuttal · Authors · 2025-03-31
>
> We thank the reviewer 1dag for the detailed review and constructive suggestions. We appreciate your acknowledgements that our **proposed metric is a reasonable choice** for facilitating theoretical analysis and that this paper is **well-written**. Below we show our reply on your comments and questions.
>
> ---
> **[R1-1]** `In Definition 3, the self-distance is nonzero. Does this indicate that Definition 3 is not suitable for measuring distances between representation spaces?`
>
> We truly appreciate this comment and would like to thank all the reviewers for pointing this out. **We found that there was a typo in Def.3** (as well as Eq.1 for recalling the definition in Guha et al. 2024) and revised Def.3 as follows. We assure you that our theoretical results and the corresponding proofs remain the same.
>
> | Submitted version: | Revised version: |
> |--|--|
> | https://hackmd.io/_uploads/ByVVMGZ6kl.png | https://hackmd.io/_uploads/Sy-BI8-Tyl.png |
>
>
> Previously, our definition of the distance between two representation spaces was based on computing the distance between the two most distant features in the respective representation spaces. Our newly revised definition only computes the maximum distance between the features of $\mathcal{R}^k_t(h_{t_1})$ and $\mathcal{R}^k_t(h_{t_2})$ with respect to the **same input**.
>
> Thus, in our revised version, the self-distance is zero.
>
> We will revise the manuscript accordingly.
>
> ---
> **[R1-2]** `Without a discussion on the tightness of the upper bound, certain interpretations—such as the peak value in Proposition 1—become questionable. `
>
>
> We thank the reviewer for the insightful comments. In response, we conducted experiments using a ReLU network on Split-CiFAR100 dataset to assess the tightness of the upper bound $U^k_t$ with respect to the representation discrepancy $D^k_t$, when $t=1$ and $k=5$; see the below plot.
>
> https://hackmd.io/_uploads/SyV_GCD6yg.png
>
> Although there exists a gap between $U^k_t$ and $D^k_t$, we observe that U^k_t consistently tracks the trend of D^k_t, suggesting that its analysis still yields meaningful insights on D^k_t.
>
> We will include this discussion in the revised manuscript.
>
> ---
> **[R1-3]** `Other aspects of the paper may also raise concerns about precision and tightness, such as Definitions 3 and 4. The proposed distance/discrepancy measures are only approximations based on imprecise beliefs (as indicated by the use of “ \approx ” in Section 4.2).`
>
> We agree with the reviewer that our distance/discrepancy measures lack formal guarantees for exactly quantifying the representation forgetting. However, similar approximations have been employed in prior work, e.g., Guha et al. (2024), which used similar measures to analyze the catastrophic forgetting, and their empirical results were well-aligned with their theoretical insights. In our case as well, the empirical trends closely align with the theoretical predictions, supporting the practical relevance of our formulations.
>
> ---
> **[R1-4]** `Figures 4 and 8 display multiple peaks, which contradicts the unique peak predicted by Proposition 1. Could this be evidence that the theoretical upper bound is not sufficiently tight?`
>
> Thank you for pointing this out.
>
> The reviewer is true that the number of peaks is different from theory (in Proposition 1) and practice (in Figures 4 and 8), but we claim that the number of peaks is not an important factor in terms of figuring out the overall trend. More importantly, both curves (theory and practice) have similar trend of exibithing two phases -- forgetting phase and saturation phase -- where the amount of forgetting significantly increases at the first phase, and the amount of forgetting deviates in a smaller range in the second phase. We will mention this in the revised manuscript.
>
>
> Regarding the tightness of our upper bound, please refer to our response in [R1-2].
>
>
> ---
> **[R1-5]** `How useful are the insights in this paper for improving the performance of continual learning?`
>
> Our analysis reveals two key findings with direct implications for improving performance in continual learning: (1) representation forgetting tends to occur more rapidly and severely in deeper layers, and (2) increasing network width mitigates the degree and speed of forgetting. These insights suggest practical strategies such as allocating more regularization or memory resources to deeper layers, or designing architectures with wider representations in critical layers to slow down forgetting.

---

### Decision · Program_Chairs · 2025-05-01

**Decision:**

Accept (poster)

**Comment:**

This paper presents a metric, which the authors term *representation discrepancy*, to measure representation forgetting in continual learning. The authors frame forgetting as a minimum alignment error between hidden layer representations through an optimal linear transformation measuring the minimum worst-case feature misalignment between different tasks at layer $k$. The manuscript provides both theoretical analysis and empirical validation of the proposed proxy for forgetting in continual learning. They derive an upper bound for the representation discrepancy measure and study its convergence rate under specific assumptions. The theoretical focus -- especially the analysis of different phases of forgetting/saturation and the dependence of forgetting on layer and depth -- of this work distinguishes it from previous research, which primarily emphasized experimental results. The paper's theoretical findings are supported by experiments on CIFAR-100 and ImageNet1K.

The reception of the work by reviewers was initially favorable, with reviewers primarily questioning:

+ The tightness of the upper bound approximation, which the authors point out in rebuttal is common to existing proxies on representation forgetting and point to their empirical evidence supporting the validity of $D^k_t$.
+ The validity of the core Assumption 1 across network architectures and CL scenarios. In rebuttal the authors again point to the alignment of their empirical results with the theoretical predictions, and additionally provide results for a ViT backbone and a domain-incremental scenario.

During the discussion phase the authors addressed the main concerns expressed by the reviewers, providing several new results supporting both the theoretical and empirical claims made in the original paper. The general consensus is that this is a valuable and novel contribution to the general problem of representation forgetting in continual learning. The recommendation is to Accept.